# Neurocranial anatomy of an enigmatic Early Devonian fish sheds light on early osteichthyan evolution

Alice M Clement[1,2,3†]*, Benedict King[1,4†], Sam Giles[5†], Brian Choo[1], Per E Ahlberg[2], Gavin C Young[6,7], John A Long[1,3]

[1]College of Science and Engineering, Flinders University, Adelaide, Australia; [2]Department of Organismal Biology, Evolutionary Biology Centre, Uppsala University, Uppsala, Sweden; [3]Department of Sciences, Museum Victoria, Melbourne, Australia; [4]Naturalis Biodiversity Center, Leiden, Netherlands; [5]Department of Earth Sciences, University of Oxford, Oxford, United Kingdom; [6]Department of Applied Mathematics, Research School of Physics & Engineering, Australian National University, Canberra, Australia; [7]Australian Museum Research Institute, Sydney, Australia

*For correspondence:
alice.clement@flinders.edu.au

[†]These authors contributed equally to this work

Competing interests: The author declares that no competing interests exist.

**Abstract** The skull of '*Ligulalepis*' from the Early Devonian of Australia (AM-F101607) has significantly expanded our knowledge of early osteichthyan anatomy, but its phylogenetic position has remained uncertain. We herein describe a second skull of '*Ligulalepis*' and present micro-CT data on both specimens to reveal novel anatomical features, including cranial endocasts. Several features previously considered to link '*Ligulalepis*' with actinopterygians are now considered generalized osteichthyan characters or of uncertain polarity. The presence of a lateral cranial canal is shown to be variable in its development between specimens. Other notable new features include the presence of a pineal foramen, the some detail of skull roof sutures, the shape of the nasal capsules, a placoderm-like hypophysial vein, and a chondrichthyan-like labyrinth system. New phylogenetic analyses place '*Ligulalepis*' as a stem osteichthyan, specifically as the sister taxon to 'psarolepids' plus crown osteichthyans. The precise position of 'psarolepids' differs between parsimony and Bayesian analyses.
DOI: https://doi.org/10.7554/eLife.34349.001

## Introduction

Some 98% of living vertebrate species belong to Osteichthyes, the 'bony fishes'. Osteichthyans are distinct from living jawless fishes (lampreys and hagfishes) and Chondrichthyes (the 'cartilaginous fishes': sharks, rays and chimaeras), and comprise Actinopterygii, or 'ray-finned' fishes (such as teleosts, bichirs and gars), and Sarcopterygii, or 'lobe-finned' fishes (lungfishes, coelacanths and tetrapods). The earliest osteichthyans are late Silurian (~425 million years ago), and from first appearance included large forms sometimes considered to be sarcopterygians (*Choo et al., 2014*; *Zhu et al., 2009*), but the interrelationships of the earliest members remain contentious (*Friedman and Brazeau, 2010*).

Despite a lack of clarity with respect to relationships, a number of significant discoveries in recent years has resulted in a clearer view of early osteichthyan anatomy. In particular, taxa such as *Youngolepis* (*Chang, 1982*), *Diabolepis* (*Chang and Yu, 1984*), *Kenichthys* (*Chang and Zhu, 1993*), *Psarolepis* (*Yu, 1998*), *Achoania* (*Zhu et al., 2001*), *Styloichthys* (*Zhu and Yu, 2002*), *Guiyu* (*Zhu et al., 2009*), *Tungsenia* (*Lu et al., 2012a2012*), *Megamastax* (*Choo et al., 2014*), *Sparalepis* (*Choo et al.,*

**eLife digest** All animals can be classified as either vertebrate (those that have a spine) or invertebrate (those that do not). About 98% of all living vertebrate species belong to a group called Osteichthyes, otherwise known as bony fish. Despite the name, this group also includes all four-limbed vertebrates – amphibians, reptiles, birds and mammals – since they evolved from prehistoric bony fish millions of years ago.

The oldest known bony fish can be traced back to around 425 million years. These ancient bony fish are all part of a sub-group called lobe-finned fish. Most modern bony fish, however, are part of a different sub-group called ray-finned fish, which can only be confidently traced back about 390 million years. A species called *Ligulalepis* was once thought to represent the oldest ray-finned fish. Scientists worked this out by examining a single *Ligulalepis* skull fossil from around 400 million years ago. However, subsequent studies have disputed its position in the evolutionary tree. So, the early evolution of bony fish remains poorly understood.

To address this, Clement, King, Giles et al. re-examined the original *Ligulalepis* skull fossil, alongside a newly discovered second skull fossil of the same species. Modern x-ray scanning techniques were used to produce detailed 3D models of both skulls and compare them to other prehistoric bony fish. This allowed Clement, King, Giles et al. to find *Ligulalepis's* exact place in the evolutionary family tree.

The experiments identified many previously unknown features of the *Ligulalepis* skull. These features suggest that this species was not a ray-finned fish; rather, it existed just before bony fish split into two sub-groups (lobe-finned and ray-finned). The analysis also suggests that *Ligulalepis* was the species most closely related to another group of fish called psarolepids. Overall, these findings clarify our understanding of the evolutionary tree of all vertebrates, including humans. Future research should continue using modern scanning techniques to uncover new information from old fossils and give further insights into the early evolution of vertebrates.

DOI: https://doi.org/10.7554/eLife.34349.002

*2017*), and *Ptyctolepis* (*Lu et al., 2017*) have provided new morphological information regarding the pattern of character acquisition in osteichthyans, in particular sarcopterygians. In contrast, the Silurian–Early Devonian record of actinopterygian evolution is poorly understood, confounded by fewer identified specimens known from typically fragmentary material. Despite the identification of Silurian sarcopterygians (but see *Lu et al., 2017*) necessitating the presence of contemporaneous actinopterygians (*Coates, 2009*), the oldest putative actinopterygian is the Lochkovian (~415 Ma) *Meemannia* (*Lu et al., 2016a*), with unequivocal ray-finned fishes such as *Cheirolepis* known only from the Eifelian-Givetian (~393 Ma) and younger deposits. This paucity of specimens may be a reflection of lower abundance and diversity of early actinopterygians compared to sarcopterygians (*Cloutier and Arratia, 2004*; *Friedman and Brazeau, 2010*).

Phylogenetic relationships among stem osteichthyans are also poorly resolved. Several taxa have been proposed to branch from the stem, but there is little consensus as to the membership or branching order. Indeed, it has been suggested that 'stem-group osteichthyans might not be recognized, even when their remains are discovered' (*Friedman and Brazeau, 2010*, pg. 38). *Dialipina* was originally diagnosed as a actinopterygian based on scale morphology (*Schultze, 1968*), but more recent analyses have resolved it either as an stem actinopterygian (*Giles et al., 2015b*; *Schultze and Cumbaa, 2001*) or stem osteichthyan (*Choo et al., 2017*; *Friedman and Brazeau, 2010*; *Giles et al., 2015c*; *Lu et al., 2016a*; *Qiao et al., 2016*). Taxa referred to as 'psarolepids' (sensu *Choo et al., 2017*) were originally placed as sarcopterygians: *Psarolepis* was initially described as a 'porolepiform-like' crown sarcopterygian (*Yu, 1998*) or either a stem osteichthyan or stem sarcopterygian (*Zhu et al., 1999*), with most subsequent analyses corroborating a stem sarcopterygian position (*Brazeau, 2009*; *Choo et al., 2017*; *Lu et al., 2016a*; *Qiao et al., 2016*; *Zhu et al., 2001*; *Zhu et al., 2009*); and *Guiyu* was deemed a stem sarcopterygian when first described (*Zhu et al., 2009*), a position subsequently supported in other analyses (*Choo et al., 2017*; *Lu et al., 2016a*). More recent analyses have recovered a stem osteichthyan position for 'psarolepids' (*Lu et al., 2017*: supported under parsimony, but not Bayesian, analyses), corroborating previous

suggestions by *Zhu et al. (1999)* and *Choo et al. (2017)*, as well as evidence from palaeohistological data (*Qu et al., 2015*). A Bayesian tip-dating approach provides no resolution regarding the phylogenetic position of *Guiyu, Achoania* and *Psarolepis* (*King et al., 2017*). The uncertainty regarding early osteichthyan relationships may be related to the dual problems of missing palaeontological data and difficulty in polarising osteichthyan characters, exacerbated by the discovery of osteichthyan-like anatomy in stem gnathostomes (e.g. *Janusiscus*: *Giles et al., 2015c*; *Entelognathus*: *Zhu et al., 2013*).

One taxon that may help to elucidate osteichthyan stem group phylogeny is the enigmatic *Ligulalepis. Ligulalepis toombsi* (*Schultze, 1968*) was erected and attributed to Actinopterygii byon the basis of isolated scales from the Early Devonian (Emsian) Taemas Limestones of the Burrinjuck area of New South Wales, Australia. A second species, *Ligulalepis yunnanensis Wang and Dong, 1989*, was erected on the basis of isolated scales from the Silurian (Ludlow) Miaokao Formation of Yunnan, China. Subsequently, other occurrences of isolated scales from Australia were attributed to the genus (*Burrow, 1994*; *Burrow, 1997*), including throughout the Bloomfield Limestone member to the Warroo Limestone member at Burrinjuck (*Basden and Young, 2001*); scales from at least the latter locality appear to belong to a single taxon (C. Burrow pers. comm to GCY). *Schultze (2016)* referred a jaw from the Early Devonian Trundle Beds of New South Wales to *Ligulalepis*, although no justification for this is given by Schultze. Histological sections through the jaw show teeth bearing acrodin, a hypermineralised tissue forming a tooth cap that is currently known only in actinopterygians (*Ørvig, 1973*).

An incomplete braincase and skull roof, AM-F101607, from the same Burrinjuck limestones, was described by *Basden et al. (2000)* as perhaps the 'most primitive osteichthyan braincase' known, emphasizing its unusual combination of morphological characters. These authors used the phylogenetic analysis of *Zhu et al. (1999)* to consider alternative placements of AM-F101607 as either a stem gnathostome, stem actinopterygian, or stem osteichthyan, the last option being the most parsimonious. Later, *Basden and Young (2001)* published a more detailed morphological description of this specimen, which they considered might rather represent a member of the actinopterygian stem. Consequently, the specimen was moved to '*Ligulalepis*' sp., and tentatively referred to *Ligulalepis toombsi*, on the basis that this was the only actinopterygian taxon known from the Emsian of southeastern Australia.

In addition to characters considered actinopterygian-like (dermal ornament, skull roof pattern and overall endocranial proportions; *Basden and Young, 2001*), the skull of '*Ligulalepis*' displayed characters found scattered across the gnathostome tree. 'Primitive' features included the presence of an eye stalk, myodomes for the attachment of oculomotor-innervated eye muscles, and an opening for the orbital artery (*Basden et al., 2000*). However, certain features were noted to bear resemblance to sarcopterygians, including the proportions of the (short and broad) telencephalic region, '*Psarolepis*-like' pit lines on the skull roof, shape of the basisphenoid and the shallow depth of the oticoccipital area (*Basden and Young, 2001*). The position of the hyomandibular attachment along the anteroposterior axis of the otic capsule was considered intermediate between the posterior placement in chondrichthyans and the anterior placement in osteichthyans, similar to that in *Acanthodes* (*Basden and Young, 2001*; *Basden et al., 2000*; *Brazeau and de Winter, 2015*; *Davis et al., 2012*).

Subsequent phylogenetic analyses have recovered contrasting placements for '*Ligulalepis*'. *Zhu et al. (2001)*, in describing the primitive sarcopterygian *Achoania* from the Early Devonian of China, resolved '*Ligulalepis*' as a basally-branching actinopterygian. *Friedman, 2007* (pg. 311) determined '*Ligulalepis*' to be a stem osteichthyan, arguing that the 'actinopterygian affinities of *Ligulalepis* . . . have relied upon characters of uncertain polarity.' Following the discovery of *Guiyu* from the late Silurian of China, '*Ligulalepis*' was recovered as a stem sarcopterygian (*Zhu et al., 2009*), whilst *Brazeau (2009)*, resolved '*Ligulalepis*' as a stem osteichthyan. *Friedman and Brazeau (2010)* examined the early osteichthyan radiation and presented a more detailed argument for stem osteichthyan affinity of '*Ligulalepis*'. Since then, '*Ligulalepis*' has been recovered as a stem osteichthyan in most analyses (*Davis et al., 2012*; *Giles et al., 2015b*; *Giles et al., 2015c*; *Giles et al., 2017*; *Long et al., 2015*; *Lu et al., 2017*; *Zhu et al., 2016*; *Zhu et al., 2013*). However, tip dated analyses employed by *King et al., 2017* place '*Ligulalepis*' within Actinopterygii with quite strong support.

Uncertainty surrounding the phylogenetic placement of '*Ligulalepis*' clearly warrants further investigation. Here, we use micro-CT scanning to reinvestigate the anatomy of the original cranium (AM-

F101607), augmented by description of a second, recently-discovered specimen (ANU V3628). The main goals of this work are: (1) to provide a revised account of the anatomy of 'Ligulalepis', including the previously unknown anterior region of the skull roof (preserved in ANU V3628); (2) to test the anatomical interpretations produced on the basis of external investigation only (*Basden and Young, 2001*; *Basden et al., 2000*); (3) to examine the effect, if any, of this new anatomical data on the phylogenetic position of 'Ligulalepis' and understanding of early osteichthyan evolution, based on a revised version of a recent phylogenetic analysis (*Lu et al., 2017*); and (4) to investigate the implications of our phylogenetic placement of AM-F101607 and ANU V3628 for the taxonomic referral of the skulls to *Ligulalepis*.

## Skull roof

Scans of AM-F101607 reveal for the first time some of the sutures between the skull roofing bones (*Figure 1*) showing a pattern different in important respects to the previous interpretation (*Basden and Young, 2001*). Viewing the scan in Drishti reveals a set of parallel bands tracing what we assume to be bone sutures in the posterior part of the skull (*Figure 1B*). Closer inspection of the scan data reveals these bands to be high-density thickenings in the basal layer of the dermal skull roof bones (*Figure 1—figure supplement 1*). These show the outline of the postparietals (of sarcopterygians; parietals of actinopterygians) and the posterior edges of the parietals (of sarcopterygians; frontals of actinopterygians) (*Figure 1B,C*). No midline suture is evident between the postparietals, but a very faint suture between the parietals is suggested. The lateral margin of the postparietal is scalloped in such a way as to provide contact faces for a series of three bones, with faint lines visible demarcating them. The most posterior bone presumably corresponds to the tabular (of sarcopterygians; supratemporal of actinopterygians). Anterior to this is a supratemporal (of sarcopterygians; intertemporal of actinopterygians), and a broad and elongate intertemporal (of sarcopterygians; dermosphenotic of actinopterygians) borders the orbit. Unfortunately, sutures cannot be visualized in the same way in ANU V3628, despite the higher scan resolution, because no high density growth bands are evident. Instead the basal layer of the skull roof dermal bone is of uniform density and thickness. This suggests that they may vary between individuals or growth phases. Further specimens of 'Ligulalepis' are required to unambiguously determine the pattern of skull roof bones in this taxon. The presence of middle and posterior pitlines, and the supraorbital canals extending to the posterior edge of the postparietals, is confirmed in ANU V3628.

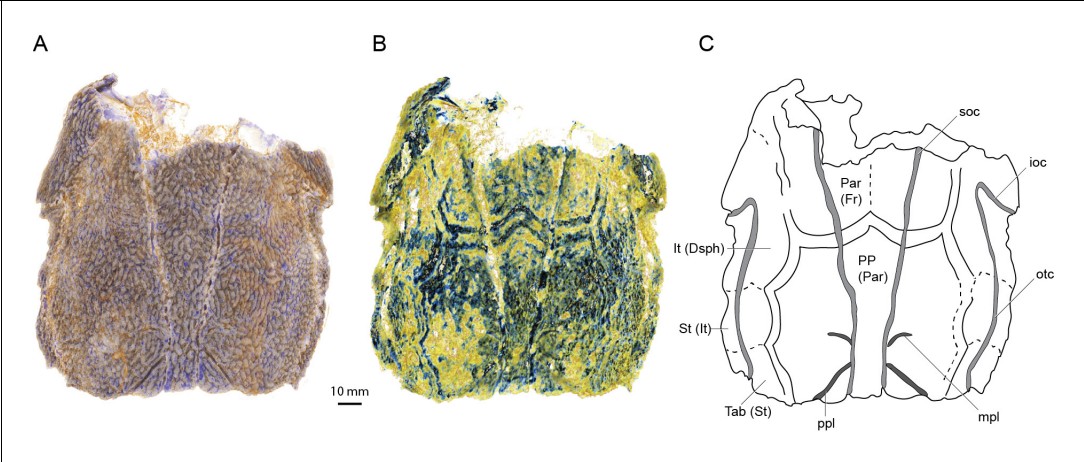

**Figure 1.** Skull roof of '*Ligulalepis*' AM-F101607 in dorsal view. Artificial colouration added in Drishti to highlight (**A**) sensory canals; and (**B**) bone sutures. (**C**) Interpretive diagram showing skull roof pattern; patterns of sensory canals inferred from both specimens. Bone names use sarcopterygian conventions, with actinopterygian conventions in brackets.
DOI: https://doi.org/10.7554/eLife.34349.003
The following figure supplement is available for figure 1:

**Figure supplement 1** Bands of high-density bone in the basal layer of the skull roof dermal bone are assumed to follow sutures in AM-F101607.
DOI: https://doi.org/10.7554/eLife.34349.004

ANU V3628 preserves the previously unknown anterior portion of the skull roof (*Figure 2*). A pineal foramen is preserved, but due to a crack in the specimen it is unclear if a separate pineal plate was present. Sutures in the anterior part of the skull are unclear. The pattern of ornamentation anterior to the pineal opening is suggestive of a median ossification (i.e. a median rostral), but this is not evident from the CT data.

The profile of the snout has a sharply downturned anterior face (*Figure 3*), as is general for gnathostomes (*Gardiner, 1984*; *Long, 1988*; *Zhu et al., 2013*; *Zhu et al., 2009*). There is an abrupt change in ornamentation on the snout, from short anteriorly directed ridges to elongate transverse ridges (*Figure 2C*). A similar pattern is known in *Dialipina* (*Schultze and Cumbaa, 2001*).

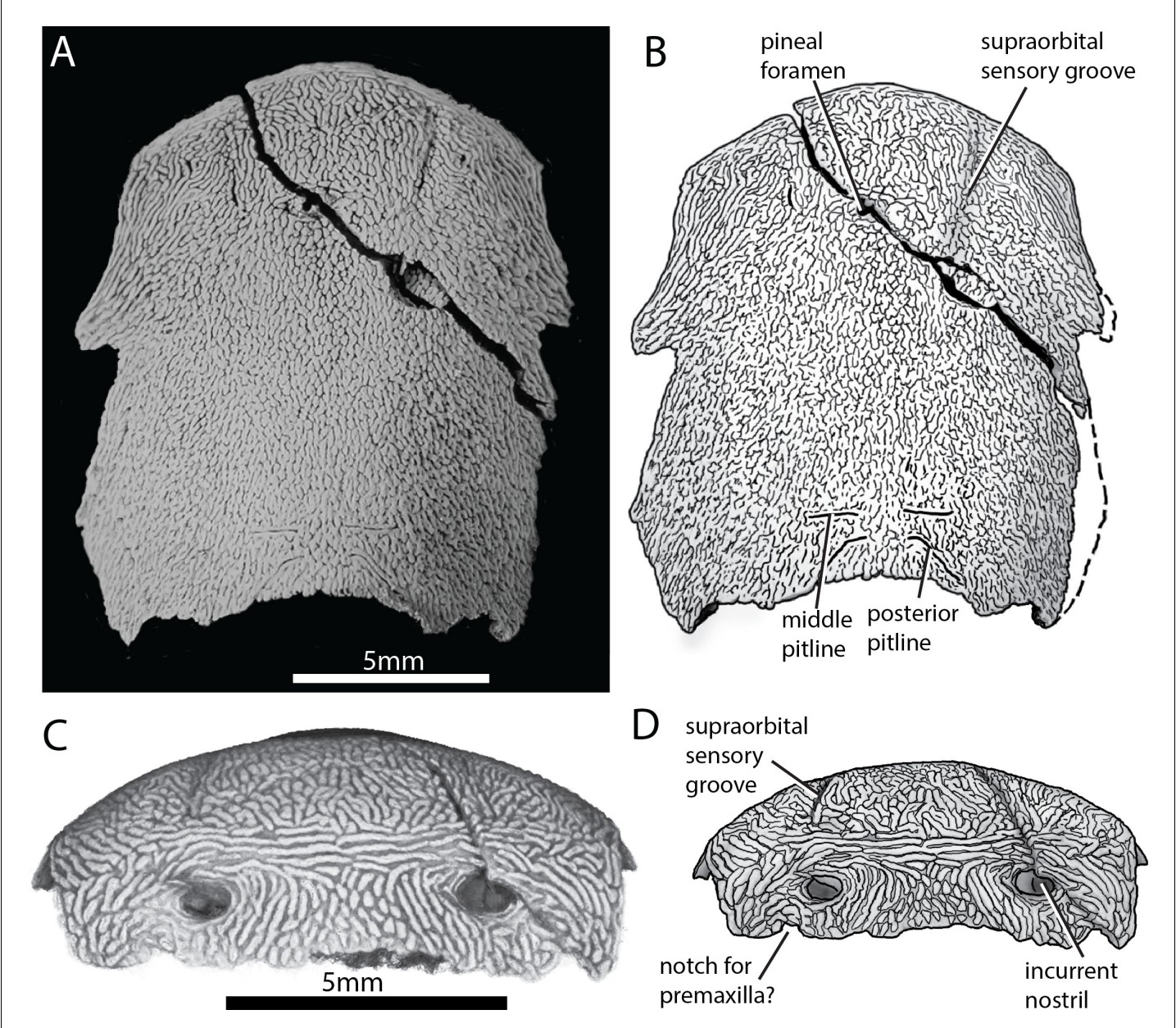

**Figure 2.** Skull of '*Ligulalepis*' ANU V3628. (**A**) Dorsal view, photograph of specimen whitened with ammonium chloride. (**B**) Line drawing of A. (**C**) Anterior view, imaged using Drishti to reveal parts embedded in resin. (**D**) Line drawing of C.
DOI: https://doi.org/10.7554/eLife.34349.005

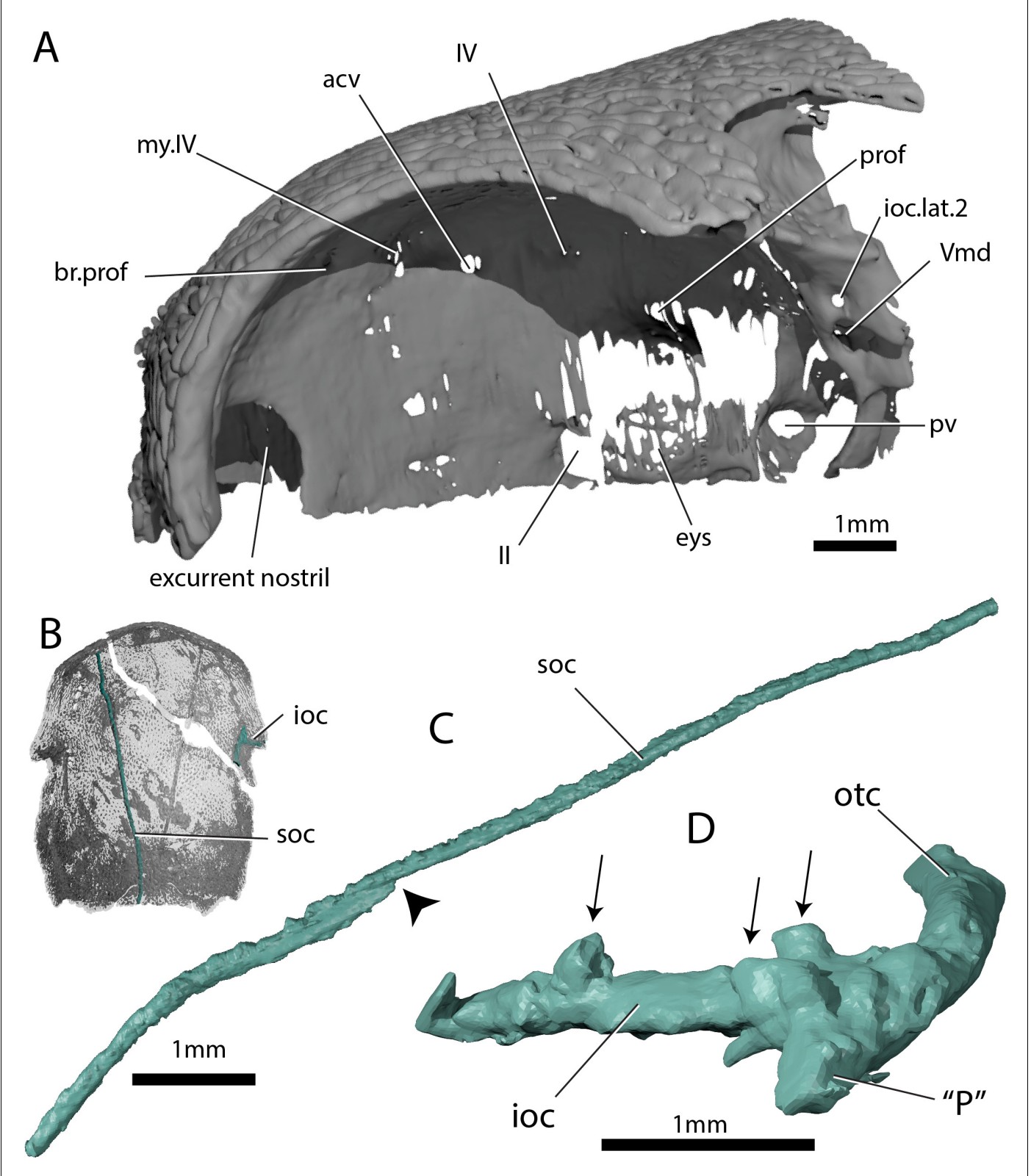

**Figure 3.** Skull and sensory canals of '*Ligulalepis*' ANU V3628. (**A**) Segmented model of dermal and perichondral bone of the left orbit, showing the posterior nostril within the orbit and endochondral bone in the eyestalk. (**B**) Position of supraorbital canal (soc) and infraorbital canal (ioc) on the skull. (**C**) Left supraorbital canal in left lateral view. Arrow indicates point where anterior and posterior canal sections overlap. (**D**) Right infraorbital and postotic canal in anterior view. Arrows indicate tubules that connect the canal to the surface.

*Figure 3 continued on next page*

*Figure 3 continued*

DOI: https://doi.org/10.7554/eLife.34349.006

The incurrent nostrils are large and widely separated from the excurrent nostrils, which appear to lie entirely within the orbits (*Figure 3*). *Basden and Young (2001)* also assumed communication of the posterior nostril with the orbit, including a notch for the nostril on the anterior margin of the orbital fenestra. Neither specimen of '*Ligulalepis*' show evidence for such a notch, although the ventral part of the nostril and orbital margin are unknown. A nostril confluent with the orbit is typically considered an actinopterygian character, but without preservation of the premaxilla and cheek bones in '*Ligulalepis*' we cannot rule out the possibility that dermal bone separated the external opening of the nostril from the orbit – for example a postero-dorsal process of the premaxilla as in *Psarolepis* (*Yu, 1998*), and perhaps *Cheirolepis* (*Gardiner, 1984*, Fig. 49). However, in '*Ligulalepis*' the opening for the posterior nostril in the endocranium lies directly within the orbit (*Figure 3A*). This is in contrast to the situation in both actinopterygians and sarcopterygians, where an endoskeletal lamina (the postnasal wall) separates the nostril and the orbit (e.g. *Gardiner, 1984*, Fig. 13). '*Ligulalepis*' lacks such a lamina, and in this respect more closely resembles some placoderms such as *Parabuchanosteus* (*Young, 1979*) and *Dicksonosteus* (*Goujet, 1984*).

## Sensory canals

The supraorbital canal (soc) extends nearly the full preserved length of the cranium, terminating a little way posterior to the downturned margin of the snout, and appears to be formed from two separate sections (*Figures 1C* and *3C*). The sections overlap slightly anterior to the level of the postorbital process, the posterior section pinching out and sitting on top of the anterior section (*Figure 3C*, indicated by arrow). Tubuli connecting the supraorbital canal to the surface are small and few in number. Tubuli connecting the infraorbital (ioc) and otic (otc) canals to the surface are larger (*Figure 3D*, arrows). The tubuli do not appear to be branched (although they may have branched in the skin above the bone), in contrast to the highly branched tubuli of some early sarcopterygians (*Bjerring, 1972*; *Clément and Ahlberg, 2010*; *Jarvik, 1972*). It is not clear whether the pores for the sensory canals figured for *Mimipiscis* and *Moythomasia* originate from branched or individual tubuli (*Gardiner, 1984*).

Anterior to the level of the pineal foramen, the supraorbital sensory canals open to the dorsal surface of the cranium (*Figure 2*), although the canal itself is housed in a ridge on the visceral surface of the skull roof. This is similar to the condition in *Achoania* (*Zhu et al., 2001*), *Guiyu* (*Zhu et al., 2009*), and *Psarolepis* (*Yu, 1998*), and may be equivalent to the 'nasal pitlines' described for *Mimipiscis* (*Gardiner, 1984*, fig. 41, 102), although in *Mimipiscis* the supraorbital canals continue anterior to the pitline. ANU V3628 is ventrally incomplete, so it is not clear if an ethmoid commissure was present. If an ethmoid commissure was present, the supraorbital canals did not communicate with it.

*Basden and Young (2001)* described a lateral notch for a preopercular sensory line, however, scans show no evidence for a preopercular canal in either specimen. A short anterior canal at the intersection of the otic and infraorbital canals is present in ANU V3628 (*Figure 3D*: 'P'), but less developed in AM-F101607. This is the 'P' canal of *Northcutt, 1989*. It is present in some acanthodians, for example *Acanthodes* (*Watson, 1937*) and some actinopterygians, namely *Mimipiscis* and *Moythomasia* (*Gardiner, 1984*). The wider distribution of the 'P' canal is difficult to assess in other taxa in the absence of exceptionally preserved material or CT data.

## Braincase

The preservation of the braincase is similar in both AM-F101607 and ANU V3628. It is mostly well ossified, and comprises the basisphenoid, orbitotemporal and otic regions. The ethmoid region is preserved on the left side of AM-F101607 and is more complete, but less well ossified, in ANU V3628. The posterior and ventral portion of the braincase, comprising the occiput and basioccipital, is absent in both specimens. The loss of this region, which is rarely preserved in early osteichthyans (e.g. *Yu, 1998*; *Zhu et al., 2001*; *Lu et al., 2012a2012*), presumably corresponds to the presence of well-developed otoccipital and ventral otic fissures, possibly in conjunction with a vestibular

fontanelle. The orbital region is large, comprising nearly half of total braincase length, and the ethmoid region short.

### Ethmoid region

The ethmoid region is very short, and is moderately well ossified. It is separated from the orbitotemporal region by a poorly developed postnasal wall. A canal leaves the cranial cavity at the left lateral limit of the pineal opening and extends posterolaterally to open into the orbit (*Figures 4C–E* and *5*, acv). This opening was identified by *Basden and Young (2001)* as for the trochlear nerve (n.IV), but its anterior position suggests it may have housed the anterior cerebral vein. This canal is present on only the left side, as in some sarcopterygians such as *Latimeria* (*Robineau, 1975*), and various

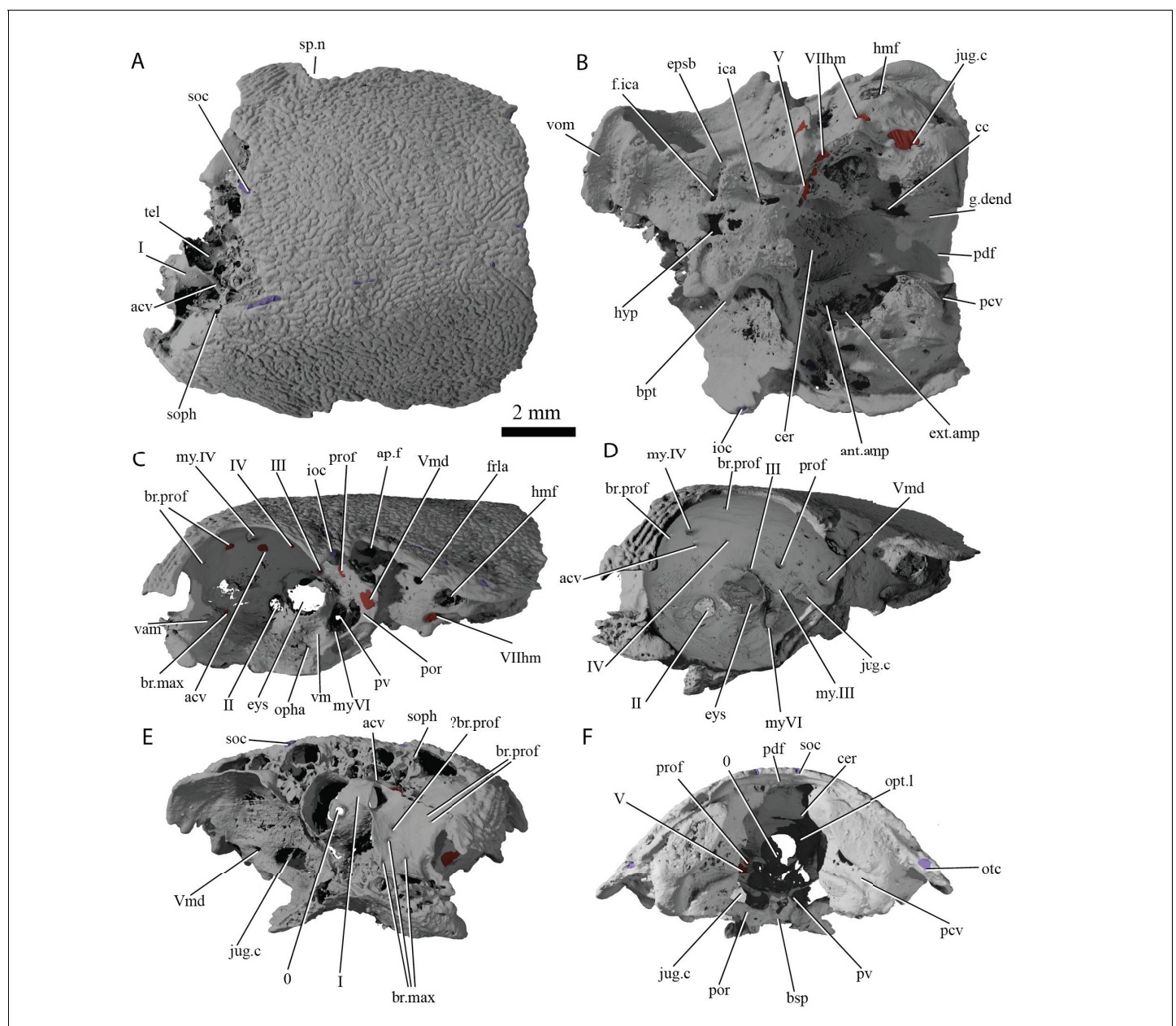

**Figure 4.** Cranium of 'Ligulalepis' AM-F101607. (A) dorsal; (B) ventral; (C) left lateral; (D) left anterolateral showing details of orbit; (E) anterior; and (F) posterior view.
DOI: https://doi.org/10.7554/eLife.34349.007

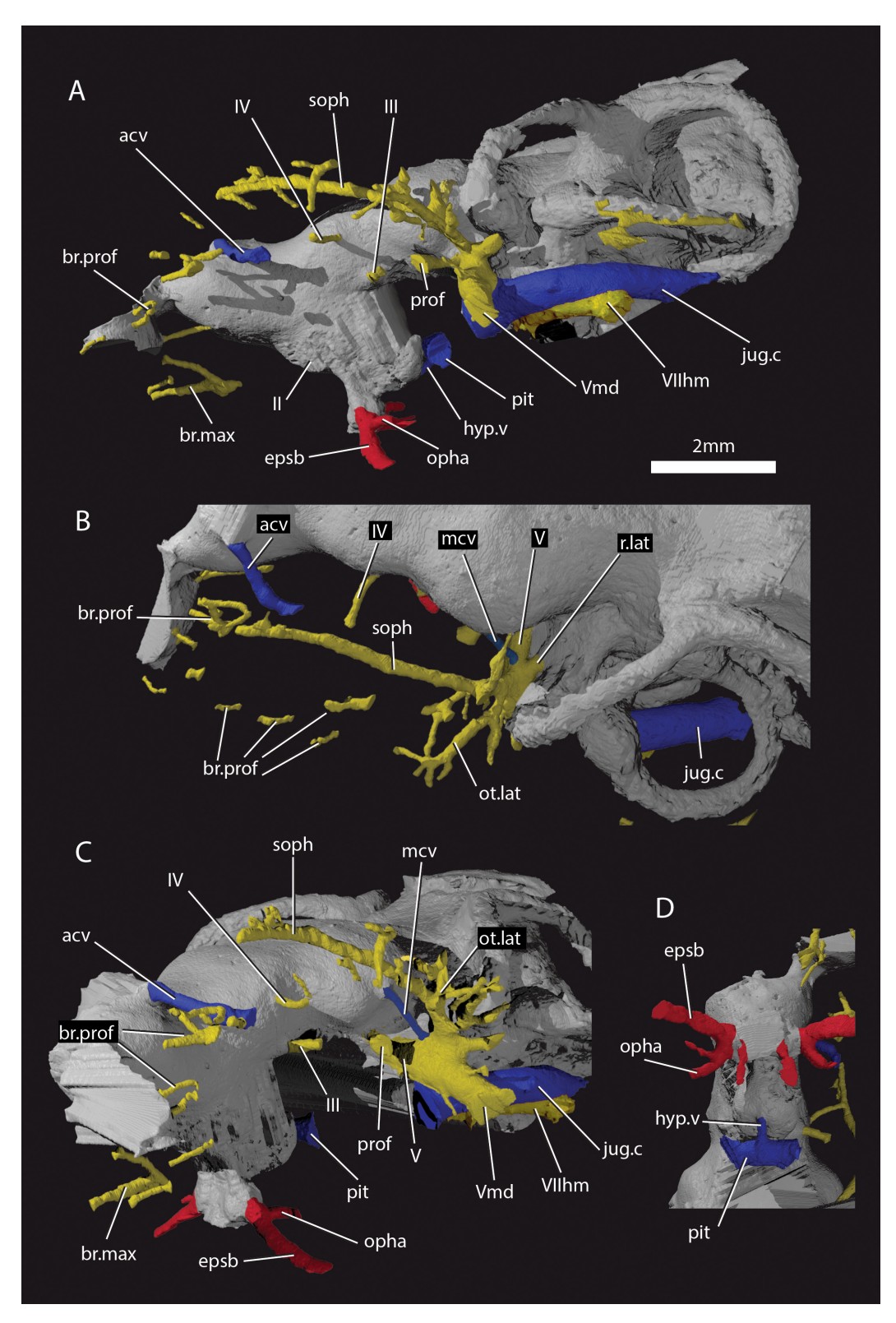

**Figure 5.** Cranial nerves and vessels of 'Ligulalepis' AM-F101607. (**A**) left lateral; (**B**) dorsal; (**C**) left anterolateral and (**D**) ventral view of anterior section only. Cranial endocast in grey, nerves in yellow, veins in blue and arteries in red.

DOI: https://doi.org/10.7554/eLife.34349.008

early actinopterygians such as *Mimipiscis* (*Giles and Friedman, 2014*) and *Kansasiella* (*Poplin, 1974*). Anterior to this, a ramifying network of canals (identified previously as for the anterior cerebral vein; *Basden and Young, 2001*: fig. 1) may have transmitted branches of the profundus nerve from the orbit to the skull roof, but their course is incomplete (*Figure 5A–C*, br.prof).

*Basden and Young (2001)* identified a number of foramina in the dorsal wall of the orbit as branches of the superficial ophthalmic nerve. However, the main trunk of the superficial ophthalmic nerve does not enter the orbit. It remains within the neurocranium, passing below the supraorbital sensory line (*Figures 5* and *6*, soph). Thus, the foramina in the orbit more likely carried branches of the profundus nerve to the skull roof (*Figures 3–6*, br.prof). The internal course of the superficial ophthalmic nerve may be related to the relatively wide interorbital septum in '*Ligulalepis*'.

Below the large opening for the olfactory canal, the posterior face of the nasal capsule is pierced by six foramina in three groups. The two dorsal-most foramina enter the nasal capsule from the orbit, and most likely transmitted branches of the profundus nerve (*Figures 4–6*, prof). The most ventral three foramina also extends from the orbit, and may have carried branches of the maxillary and buccal nerves (*Figures 4–6*, br.max). As noted by *Basden and Young (2001)*, the remaining canal originates in the forebrain, but its purpose is unclear. CT scans show that the apparent foramen at the anterior extent of the basisphenoid (*Basden and Young, 2001*: fig.7, ?fica) is in fact blind.

## Orbitotemporal region

The orbitotemporal region is extensive, forming the widest part of the braincase and comprising nearly half the total length. A large opening in the orbital wall of AM-F101607 represents the eyestalk attachment area, and was recognized as such due to its everted rims (*Basden et al., 2000*).

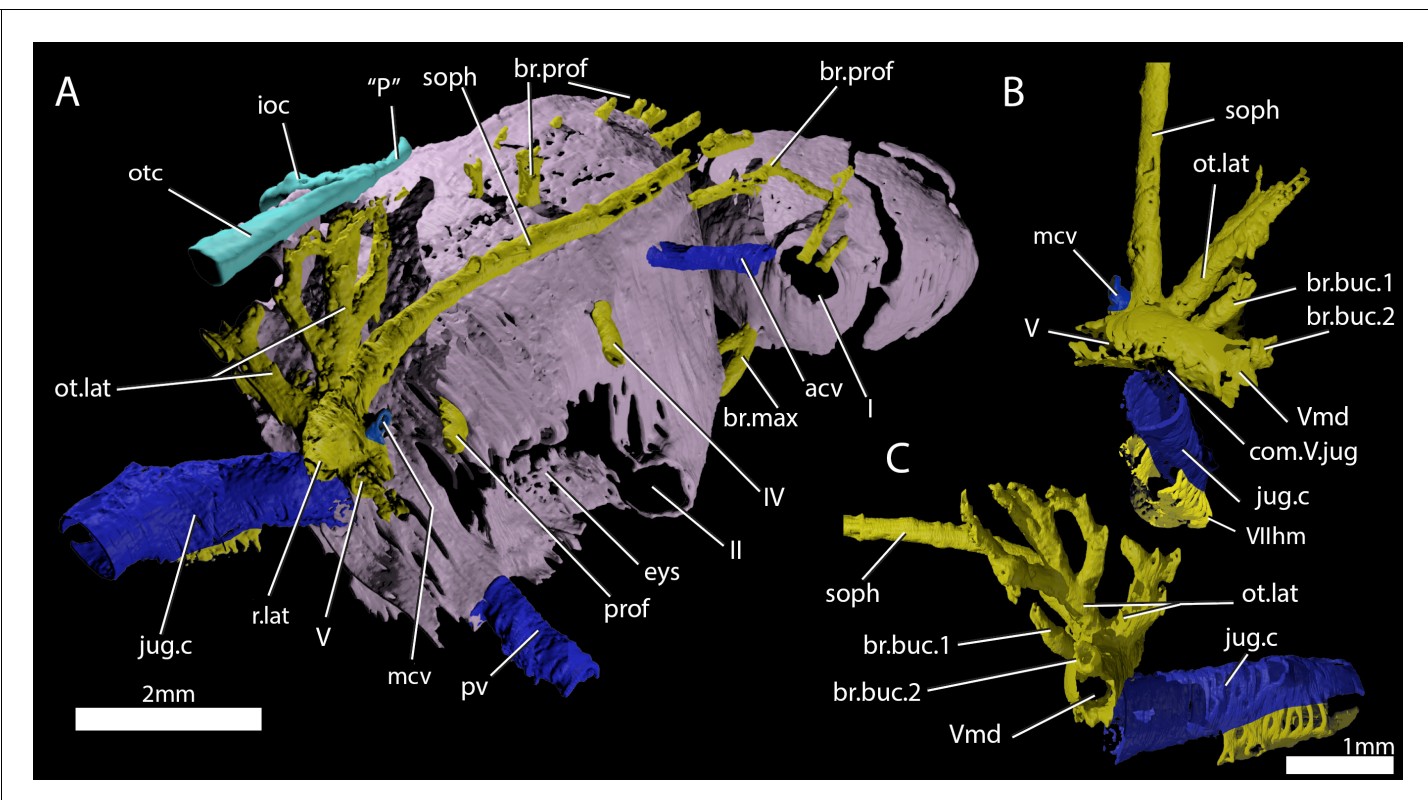

**Figure 6.** Cranial nerves and vessels of '*Ligulalepis*' ANU V3628. (**A**) ANU V3628, segmentation of the interior of the left orbital region, viewed from a postero-dorsal-medial viewpoint. The cranial endocast is not shown. Perichondral bone lining the orbit and nasal capsules is in lilac. Nerves are yellow, veins blue and sensory canals are in turquoise. The trigeminal, lateralis and facial nerves and their branches and the jugular vein, viewed from an anterior-ventral (**B**) and left lateral (**C**) viewpoints.

DOI: https://doi.org/10.7554/eLife.34349.009

The scan of ANU V3628 reveals delicate endochondral bone laminae filling the eyestalk attachment area (*Figure 3A*, eys), forming a rough surface as seen on other articular surfaces in many osteichthyans. This ossification may have been present in AM-F101607 but lost during preparation (the area is protected by resin in ANU V3628), or ANU V3628 could reflect a more advanced stage of ossification.

The oculomotor (III) and profundus (prof) nerves, as well as the entry of the jugular canal (jug.c) into the orbit, were correctly identified by *Basden and Young ([2001]*: fig. 2), although there is no communication between the profundus nerve and the canal described by *Basden and Young (2001)* as housing the orbital artery. The position of the pituitary vein and ophthalmic artery foramina can also be confirmed (*Figures 4C* and *5A,C,D*, pv, opha).

The pituitary vein is continuous between the orbits, and is connected to the hypophysial chamber by a median hypophysial vein (*Figure 5D*, hyp.v). This condition is similar to the transverse pituitary vein in placoderms, for example *Brindabellaspis* (*Young, 1980*), *Parabuchanosteus* (*Young, 1979*), *Jagorina* (*Stensiö, 1969*) and probably *Romundina* (*Dupret et al., 2017*; *Goujet and Young, 2004*). Petalichthyid placoderms however lack this character (*Castiello and Brazeau, 2018*). In some early diverging arthrodire placoderms (e.g. *Kujdanowiaspis, Dicksonosteus*), the pituitary vein is continuous, but exits the floor of the braincase via the subpituitary fossa. There is no foramen in the hypophysial fossa identified that could have carried a median hypophysial vein in these taxa (*Goujet, 1984*; *Stensiö, 1963b*). The condition in '*Ligulalepis*' contrasts with other crown gnathostomes in which the pituitary vein enters the hypophysial chamber directly (e.g. *Maisey, 2005*; *Maisey, 2007*; *Holland, 2014*; *Giles and Friedman, 2014*). A continuous transverse pituitary vein canal may be partly linked to the relative position of the forebrain and the angle of the hypophysial chamber in '*Ligulalepis*'.

The trochlear (IV) nerve enters the orbit dorsal to the eyestalk attachment area, some way posterior to the dorsal myodome (my.IV) and anterior cerebral vein (*Figures 3* and *4C,D*). The canal originally identified for the trochlear (IV) nerve (*Basden and Young, 2001*) in fact houses the anterior cerebral vein. This revised position of the trochlear (IV) nerve (i.e. posterodorsal to the eyestalk) reflects the general gnathostome condition (*Chang, 1982*; *Gardiner, 1984*; *Maisey, 2005*; *Young, 1979*), but the position of the myodome anterior to the orbit is more similar to that of osteichthyans.

A large opening on the postorbital process was previously identified (*Basden and Young, 2001*) as housing the orbital artery, in line with the position of this feature in placoderms. Segmentation of the internal course of this canal shows there is no communication with the profundus nerve canal, contrary to *Basden and Young (2001)* description. The canal connects with a large opening beneath the cerebellar portion of the cranial cavity, most parsimoniously identified as the root of the trigeminal (V) nerve. As such, the large foramen in the orbit most likely transmitted the mandibular branch of the trigeminal nerve (*Figures 4C–E* and *5A,C*, Vmd). This canal also aligns with a notch in the postorbital process, along which the mandibular nerve would have travelled. This morphology is similar to that seen in chondrichthyans (e.g. *Cladodoides*: *Maisey, 2005*; '*Cobelodus*': *Maisey, 2007*). Small branches are given off the trigeminal nerve within the braincase. One branch (*Figure 6B–C*, br. buc.1) enters the posterodorsal part of the orbit at a steep angle and likely carried lateralis fibres to small canals in the roof of the orbit that lead to the dorsal part of the infraorbital canal. A second branch (*Figure 6B–C*, br.buc.2), previously suggested as carrying the posterior branch of the oculomotor (III) nerve (*Basden and Young, 2001*), opens onto the postorbital process just dorsal to the opening for the mandibular branch. This may also have carried lateralis fibres to the infraorbital canal.

Posterior to the root of the trigeminal nerve, a canal (r.lat) leaves the anterior face of the utricular region and enters the "trigemino-facialis chamber" (*Figure 5B*). This is interpreted as the root of the anterior lateral line nerves, in a similar position as in other early osteichthyans (*Jarvik, 1980*; *Chang, 1982*; *Giles and Friedman, 2014*). An additional canal (mcv) exits the cranial cavity from the midpoint of the cerebellum and enters the "trigemino-facialis chamber" at a steep angle (*Figure 5C*). Due to its position and orientation, this is interpreted as the middle cerebral vein. The jugular canal communicates with the "trigemino-facialis chamber" via an opening in the roof of the canal (*Figure 6B*, com.V.jug), through which the middle cerebral vein and the maxillary branch of the trigeminal nerve may have been transmitted (*Basden and Young, 2001*).

The identity of the large foramen in the dorsal portion of the anterior pocket (*Figure 4C*, ap.f) is hard to discern. Segmentation reveals a cavity that is continuous ventrally and dorsally with the infraorbital canal, and may be related to the spiracle. The cavity is also connected with the otic nerve anteriorly. The openings identified by *Basden and Young, ([2001]*: fig 2) ventral to this foramen do not continue within the bone.

Further clarifications can be made to the identity of the large foramina on the lateral and ventral face of the otic region (*Figure 4*). The canal ventral to the hyomandibular facet intersects the ventral portion of the jugular canal and can be traced to the ventral otic fissure. It can be confirmed as the hyomandibular trunk of the facial nerve (VIIhm; *Basden and Young, 2001*: figs. 2,3).

*Young, 1979Dupret et al., 2017Maisey, 2005Maisey, 2007Maisey, 2005Maisey, 2007*

The foramen identified by *Basden and Young ([2001]*: figs 2,3) as for the glossopharyngeal nerve is in fact the posterior exit of the jugular canal (*Figure 4B,F*, jug.c); the glossopharyngeal nerve presumably exited through the otic-occipital fissure. *Giles et al., 2015c*

### Ventral surface

As outlined by *Basden and Young (2001)*, the internal carotids enter the braincase through two foramina flanking the median hypophysial opening before giving off the efferent pseudobranchial and ophthalmic artery (*Figure 4B*, epsb, *Figure 4C*, opha). As in chondrichthyans (*Maisey, 2005*; *Maisey, 2007*), but unlike in osteichthyans (*Chang, 1982*; *Gardiner, 1984*) and placoderms (*Hu et al., 2017*; *Young, 1980*), there is no evidence of a parabasal canal carrying the palatine artery anterior to this point. *Basden and Young (2001)* identified grooves on the ventral surface of the basisphenoid as for the lateral dorsal aorta. However, since these grooves are anterior to the efferent hyoid artery we prefer to refer to them as the internal carotid arteries (*Figure 4B*, ica). Although (*Basden and Young [2001]* : fig. 3) identified foramina for the palatine branch of the facial nerve and the orbital artery in the roof of the canal for the internal carotid (their lateral dorsal aorta), the roof appears to be complete.

## Cranial endocast

A comparison of the two cranial endocasts is shown in *Figure 7*. Differences in appearance largely relate to the presence of extensive rock matrix surrounding ANU V3628, in contrast to the acid-prepared cranium of AM-F101607. The external walls of the endocranial cavity are largely complete in both specimens, although as the parachordal plate of the braincase is not preserved the ventral extent is uncertain. Overall, the endocast of '*Ligulalepis*' is short and broad, particularly the otic region (*Figure 7A,B*). The proportions occupied by different regions are similar to early chondrichthyans, with the forebrain section comprising less than 20% of the total length, the midbrain section around 15%, and the hindbrain section some 65%.

Description of the endocast allows the identity of features within the cranial cavity to be revised. A distinct depression in the roof of the cranial cavity, medial to the otic capsule, was considered by (*Zhu et al. ([2010]*, fig. 4c) to be evidence of a lateral cranial canal. This embayment is in fact the crus commune of the anterior and posterior semicircular canal (*Figure 4B*). The groove anterior to this is somewhat shallower in the braincase and indicates where the roof of the utricular region joins the rest of the cranial cavity (the groove for anterior and posterior semicircular canals of *Basden and Young 2001*: fig. 3).

### Forebrain

The region of the endocast corresponding to the forebrain comprises space for the olfactory bulbs, telencephalon and diencephalon. This region in '*Ligulalepis*' is relatively wide (*Figure 7A,B*), comparable to the forebrain in placoderms such as *Macropetalichthys* (*Stensiö, 1963a*) and chondrichthyans such as *Orthacanthus* (*Schaeffer, 1981*). However, it is still only half the width of the cerebellum. The short, wide olfactory tracts leave the anterolateral corners of the telencephalic region in separate tracts and connect to the bulbous nasal capsules, preserved in ANU V3628 (*Figure 7B*, n.cap). The short olfactory tracts are similar to those of placoderms, for example *Buchanosteus* (*Young, 1979*) and *Kujdanowiaspis* (*Stensiö, 1963a*), as well as chondrichthyans such as *Cladodoides* (*Maisey, 2005*) and *Orthacanthus* (*Schaeffer, 1981*), but also some sarcopterygians such as *Tungsenia* (*Lu et al., 2012a2012*) and *Qingmenodus* (*Lu et al., 2016*). A small canal for the

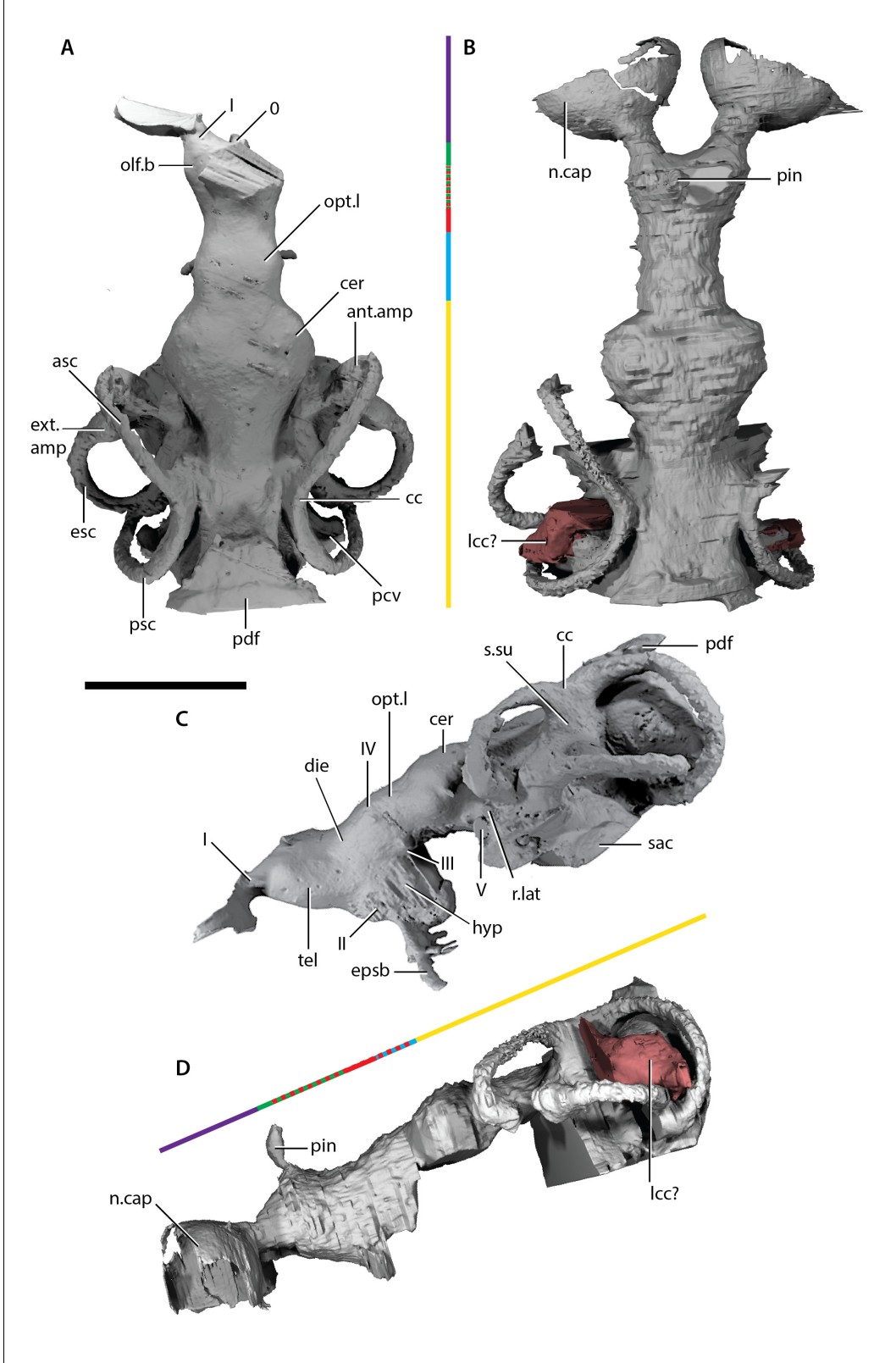

**Figure 7.** Endocast of 'Ligulalepis'. (**A**) AM-F101607, dorsal view. (**B**) ANU V3628, dorsal view. (**C**) AM-F101607 lateral view. (**D**) ANU V3628 lateral view with possible lateral cranial canal in red. Major brain regions indicated by coloured bars: nasal capsules (purple), telencephalon (green), diencephalon (red), mesencephalon (blue), metencephalon and myelencephalon (yellow).
DOI: https://doi.org/10.7554/eLife.34349.010

terminal nerve (0) exits from the anterior face of the forebrain, between the olfactory tracts, in AM-F101607.

The telencephalon is the widest and highest portion of the forebrain. At its widest point it measures 6 mm in AM-F101607. It is developed into slight lobes dorsolaterally; these may represent olfactory bulbs (*Figure 7A*, olf.b). The dorsal roof of this region is preserved in ANU V3628, as is the canal to the pineal opening (*Figure 7B,D*, pin). The oblique crack across ANU V3628 intersects the pineal opening, so it is unclear whether or not a parapineal organ was present. The margin between the telencephalic and diencephalic regions is marked by a gentle constriction in the endocranial cavity.

The region of the endocast corresponding to the diencephalon is short and narrow in dorsal view. Ventrally, the diencephalic region extends to the floor of the cranial cavity, being continuous with the hypophyseal fossa (hyp), and projects posteriorly some way under the mesencephalon (*Figure 7C*). This region is unfinished posteriorly, and it is unclear whether a saccus vasculosus was present as in actinopterygians (*Giles and Friedman, 2014*). The lateral wall of the diencephalic region of the endocast is unfinished for the eyestalk attachment area in AM-F101607 (*Figure 4C,D*; this area is not preserved in ANU V3628). The optic nerves (II) enter the orbit through a large foramen at the anterolateral limit of the diencephalon (*Figures 6A* and *7C*). Beneath this opening, a vertical ridge on the side of the hypophysial chamber likely shows the course of the internal carotid artery after it enters the braincase. The efferent pseudobranchial artery joins the internal carotid at the point of entry into the braincase (*Figure 4B*, epsb, f.ica), and internally the ophthalmic artery branches from the same point and enters the orbit (*Figure 5D*, opha). The hypophysis is oriented ventrally, in agreement with the generalised osteichthyan condition (e.g. *Youngolepis*: *Chang, 1982*; *Mimipiscis*: *Giles and Friedman, 2014*), but unlike the posterodorsally-oriented hypophysis seen in *Cladodoides* (*Maisey, 2005*).

## Midbrain

Posterior to the diencephalic portion of the endocast, the region corresponding to the midbrain (mesencephalon) widens slightly in AM-F101607. The midbrain cavity is not differentiated into separate recesses for each optic lobe (*Figure 7A*, opt.l), which appears to be the general gnathostome condition. There are similarly slight bulges in chondrichthyans (e.g. *Cladodoides, Xenacanthus*), whereas highly distinct optic lobes are seen in actinopterygians crownward of *Mimipiscis* (*Coates, 1999*; *Giles and Friedman, 2014*). A narrow, dorsally positioned canal leaves the cranial cavity and enters the orbit (*Figure 5A–C*, IV). This foramen was illustrated, but not identified, by *Basden and Young [2001]*: fig. 2b, the opening posterior to that labeled IV and dorsal to the eyestalk). The position of the canal strongly suggests it housed the trochlear nerve (IV), given a similar placement in crown gnathostomes (*Chang, 1982*; *Giles and Friedman, 2014*; *Maisey, 2005*). More ventrally, the oculomotor (III) nerve leaves the midbrain and enters the orbit (*Figure 5A,C*); there is no evidence that this nerve bifurcated along its course. The oculomotor nerve does not typically bifurcate in chondrichthyans (e.g. *Cladodoides*, *Maisey, 2005*) or sarcopterygians (e.g. *Eusthenopteron*, *Jarvik 1980*; *Youngolepis*, *Chang 1982*, ), and is variably developed in actinopterygians such as *Mimipiscis* (*Giles and Friedman, 2014*) and *Lawrenciella* (*Hamel and Poplin, 2008*).

## Hindbrain

The hindbrain is the widest portion of the endocast and would have housed the metencephalic and myelencephalic brain regions in life. The cerebellum extends anterior to the labyrinth (*Figure 7A,B*, cer), as in chondrichthyans (e.g. *Cladodoides*, *Maisey, 2005*) and, to a lesser extent, sarcopterygians (e.g. *Eusthenopteron*, *Jarvik 1980*). Although the dorsal surface bears a slight suggestion of two lobes, these can hardly be compared to the distinct cerebellar auricles of actinopterygians such as *Mimipiscis* (*Giles and Friedman, 2014*). Similarly, there is no obvious protrusion housing the cerebellum corpus.

The profundus nerve (prof) leaves the cranial cavity separately from the trigeminal nerve (V) and enters the orbit (*Figure 6A*). *Northcutt and Bemis, 1993* made a case that the profundus should be considered a separate nerve rather than a branch of the trigeminal, based on developmental evidence and the separation of these nerves in chondrichthyans, early actinopterygians and *Latimeria*. The cranial nerve configuration seen in '*Ligulalepis*' adds to a growing body of evidence from fossil

endocranial studies that the separation of the trigeminal and profundus nerves is indeed the plesiomorphic state for crown gnathostomes (*Chang, 1982*; *Giles and Friedman, 2014*; *Maisey, 2005*).

Posterior to the cerebellum, the dorsal part of the hindbrain narrows and drops in height, before the endocast broadens again at the midpoint of the labyrinth. The entire dorsal surface of the hindbrain is smooth, and does not rise as high dorsally as the crus commune of the anterior and posterior semicircular canals (*Figure 7C*, cc). The posterior dorsal fontanelle is trapezoidal in outline (*Figure 7A*, pdf). A ridge on the dorsal surface at the lateral edge of the hindbrain may indicate the path of the endolymphatic ducts within the cranial cavity into the posterior dorsal fontanelle (*Figure 4B*, g.dend).

ANU V3628 appears to have a lateral cranial canal (*Figure 7B,D*, lcc?), as in actinopterygians (*Giles et al., 2018*). *Basden and Young ([2001]*: Fig. 3) identified a groove for the posterior cerebral vein in AM-F101607, in a corresponding position to a similar groove in *Mimipiscis* and *Moythomasia* (*Gardiner, 1984*). In ANU V3628, the dorsal part of this groove contains a large foramen, most clearly developed on the left side (*Figure 8C,D*), in the same position to the opening for the lateral cranial canal in *Moythomasia* (*Gardiner, 1984*, fig. 27) and *Mimipiscis* (*Gardiner, 1984*, fig. 11). Segmentation reveals that this foramen opens into a large unossified space (*Figure 8E*), as expected for a lateral cranial canal (*Gardiner, 1984*; *Patterson, 1975*), although this cavity does not have a continuous perichondral lining.

However, the situation regarding a lateral cranial canal in AM-F101607 is less clear. *Basden and Young (2001)* identified foramina in the posterior cerebral vein groove, and identified them as anterior tributaries of the posterior cerebral vein. Although the foramina on the left hand side are indeed small (*Figure 8B*), on the right hand side there is a larger, more distinct foramen in the same position (*Figure 8A*). However, there is no obvious connection between the unossified space and the dorsal part of the lateral endocranial wall, and furthermore this cavity appears interconnected with much of the remaining interperichondral space in the otic region of the braincase. The lateral cranial canal may have been variable in its development, as has been suggested for *Mimipiscis* (*Gardiner, 1984*). It seems likely, however, that the foramina in ANU V3628 are far too large to be identified as tributaries of the posterior cerebral vein. As the occipital portion of the braincase is missing, the posterior extent of the hindbrain cannot be described.

## Labyrinth

The labyrinth region in '*Ligulalepis*' is well preserved in AM-F101607 (*Figure 7A,C*), with three complete, slender semicircular canals present, and all carrying small expansions for ampullae.

The anterior semicircular canal (asc) is anteroposteriorly long, but does not extend far ventrally. In contrast, the posterior semicircular canal (psc) is tall dorsoventrally, but anteroposteriorly very short. A short portion of preampullary canal separates the posterior ampulla from the cranial cavity. The posterior semicircular canal curves back underneath the external semicircular canal (esc) to meet the cranial cavity far ventrally. This ventral position of the posterior canal is reminiscent of that in placoderms (e.g. *Dicksonosteus*; *Goujet, 1984*), chondrichthyans (*Schaeffer, 1981*; *Maisey, 2007*), early sarcopterygians (e.g. *Youngolepis*; *Chang, 1982*) and, to a slightly lesser extent, in the early actinopterygian *Mimipiscis* (*Giles and Friedman, 2014*).

Strikingly, the external canal is positioned obliquely at an angle of about 30 degrees from the cranial cavity, and completes nearly a full circle before re-entering the vestibule (*Figure 7A,C*). The posterior connection with the cranial cavity is swollen, almost giving the appearance of an ampulla like that at the anterior extent of the canal (ext.amp).

Other notable features of the vestibular system are the relatively shallow superior sinus (*Figure 7C*, s.su) situated below the crus commune, seen elsewhere in *Cladodoides*, *Youngolepis* and *Kansasiella* (*Chang, 1982*; *Maisey, 2005*; *Poplin, 1974*), but not in *Mimipiscis* (*Giles and Friedman, 2014*) or *Acanthodes* (*Davis et al., 2012*). As well as the crus commune, a portion of the sinus superior, anterior and posterior semicircular canals project dorsally above the endocranial roof. The same condition is found in chondrichthyans and early actinopterygians (*Giles and Friedman, 2014*).

Although incompletely known ventrally, the sacculus is not laterally extensive and appears to have been shallow (*Figure 7C*, sac). The general morphology of the labyrinth, including the dorsoventrally extensive posterior canal, which projects above the endocranial roof as well as below the cerebellar floor, and the inclined external canal, recalls that of an early chondrichthyan such as

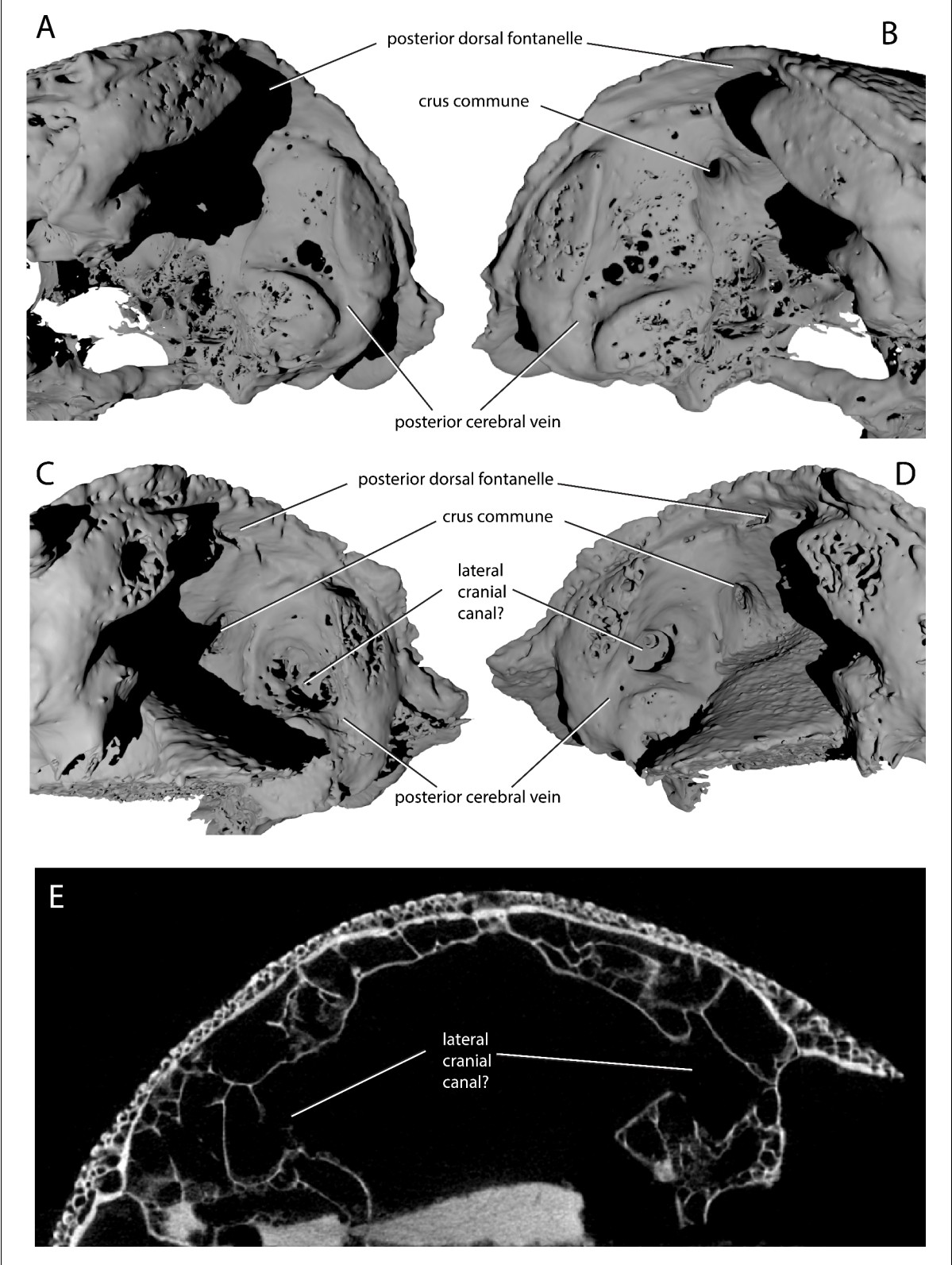

**Figure 8.** Variability in the development of a lateral cranial canal in '*Ligulalepis*'. (A–B) Ventrolateral view of AM-F101607, showing internal view of the otic region on the right hand side (A) and the left hand side (B). (C–D) Ventrolateral view of ANU V3628, showing internal view of the otic region on the right hand side (C) and the left hand side (D). (E) CT scan cross-section of ANU V3628 showing diverticula that may represent lateral cranial canals.
DOI: https://doi.org/10.7554/eLife.34349.011

*Cladodoides* (*Maisey, 2005*) or perhaps even *Acanthodes* (*Davis et al., 2012*). The labyrinth is far removed from that seen in *Mimipiscis* (*Giles and Friedman, 2014*), or sarcopterygians such as *Eusthenopteron* and Devonian lungfishes (*Clement and Ahlberg, 2014*; *Clement et al., 2016*; *Jarvik, 1980*). No otoliths were recovered from the specimen, although their absence is most likely due to either a failure to be preserved or a consequence of acid preparation.

A life reconstruction of '*Ligulalepis*' based on the skull morphology of AM-F101607 and ANU V3628 (other features remain hypothetical) is presented in *Figure 9*.

## Phylogenetic analysis

AM-F101607 and ANU V3628 were coded into an updated phylogenetic analysis modified from *Lu et al., 2017*. As well as changes to anatomical scores for '*Ligulalepis*', codes for several taxa were updated and some characters were deleted or reformulated to give a total of 282 characters coded for 94 taxa (for full details see the 'phylogenetic methods' section). This dataset was analysed using both parsimony and Bayesian inference. The parsimony analysis retrieves *Dialipina*, '*Ligulalepis*', and 'psarolepids' as successively branching sister taxa to the osteichthyan crown node (*Figure 10A*). However, support for the clade that comprises crown osteichthyans (as retrieved from this analysis) is low, with Bremer support of 1 and a bootstrap of just 4. This is very weak support, although we note that bootstrap values obtained from TNT are likely to be much more conservative than those produced by PAUP*: bootstrap values in TNT are calculated from the strict consensus trees found in each replicate (*Goloboff et al., 2008*), whereas PAUP* uses all the shortest trees from each replicate, weighted by the reciprocal of the number of trees found in that replicate (*Swofford, 2003*).

There are six unambiguous character state changes on the branch leading to crown osteichthyans. These are #78 (enameloid on teeth gained), #110 (shape of parashenoid splint shaped), #116 (olfactory tracts long), #130 (eyestalk absent), #184 (median dorsal plate absent), #211 (dorsal fin spines absent). Of these, only the olfactory tracts and eyestalk are known in '*Ligulalepis*'.

Alternative phylogenetic placements under parsimony were tested using two constrained searches, one with '*Ligulalepis*' constrained within actinopterygians and another with 'psarolepids' constrained within sarcopterygians. A stem actinopterygian position for '*Ligulalepis*' requires a single additional step, and the grouping of '*Ligulalepis*' and actinopterygians was found in 18% of the bootstrap replicates. Enforcing this topology also resulted in 'psarolepids' being resolved as stem sarcopterygians (*Figure 10B*). A single additional step is required to place 'psarolepids' on the sarcopterygian stem, and this grouping is found in 16% of bootstrap replicates. When this grouping is enforced it leads to '*Ligulalepis*' falling into a polytomy with actinopterygians and sarcopterygians (*Figure 10B*).

The Bayesian analysis retrieves 'psarolepids' on the sarcopterygian stem with moderately strong support (pp = 0.94, *Figure 11*). '*Ligulalepis*' is resolved as a stem osteichthyan in the 50% majority rule tree (*Figure 11*), although the crown osteichthyan clade has weak support (0.61). However, an actinopterygian position for '*Ligulalepis*' has a posterior probability of 0.22.

## Discussion

### '*Ligulalepis*' and early osteichthyan phylogeny

'*Ligulalepis*' is recovered as a stem osteichthyan in the phylogenetic analysis, specifically as the sister lineage to 'psarolepids' (*Guiyu, Sparalepis, Psarolepis, Achoania*) plus crown Osteichthyes (*Figures 9* and *10*). *Dialipina* is resolved as the sister taxon to all other osteichthyans. However, the placement of '*Ligulalepis*' as the earliest diverging stem actinopterygian requires only a single additional step, and evidence for an actinopterygian affinity must be considered. Cranial features previously suggested as linking '*Ligulalepis*' with actinopterygians (*Basden and Young, 2001*) are now better considered to be general osteichthyan characters (e.g. dermal ornament) or of uncertain polarity (skull roof pattern and overall structure). Of the three characters proposed by *Lu et al., 2017* as uniting ray-finned fishes inclusive of *Meemannia*, '*Ligulalepis*' lacks two: posteriorly expanded tabulars (supratemporals of actinopterygians) and a spiracular canal. The remaining character, presence of a lateral cranial canal (*Coates, 1999*; *Gardiner, 1984*), is harder to assess. Primitively, the lateral cranial canal connects with the endocavity through the loop of the posterior semicircular canal, but in neopterygians it may connect with the cranial cavity anteriorly (e.g. '*Caturus*', *Rayner, 1948*,

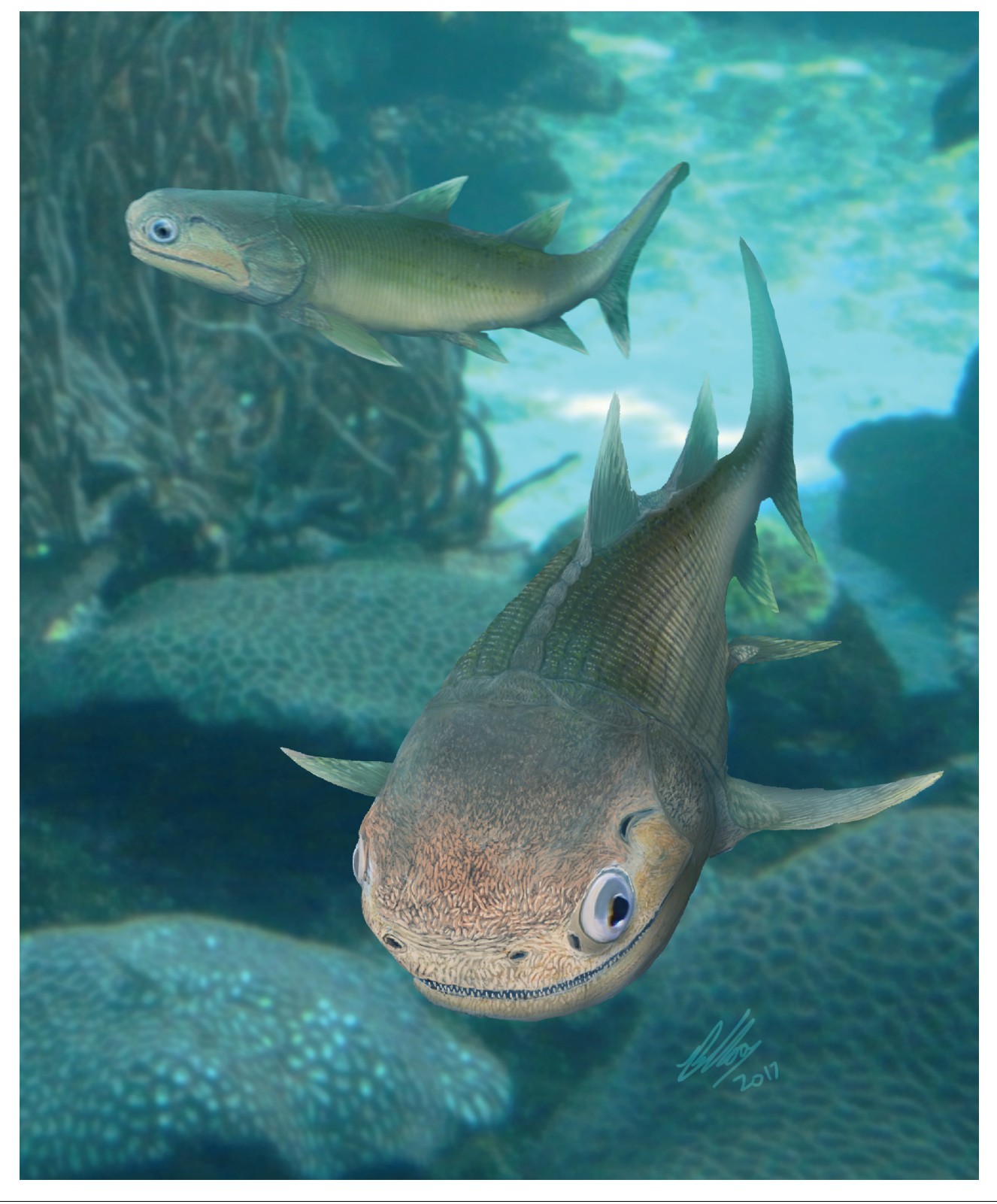

**Figure 9.** Life reconstruction of 'Ligulalepis'. Based on the skull roof morphology of AM-F101607 and ANU V3628, other features remain hypothetical.
DOI: https://doi.org/10.7554/eLife.34349.012

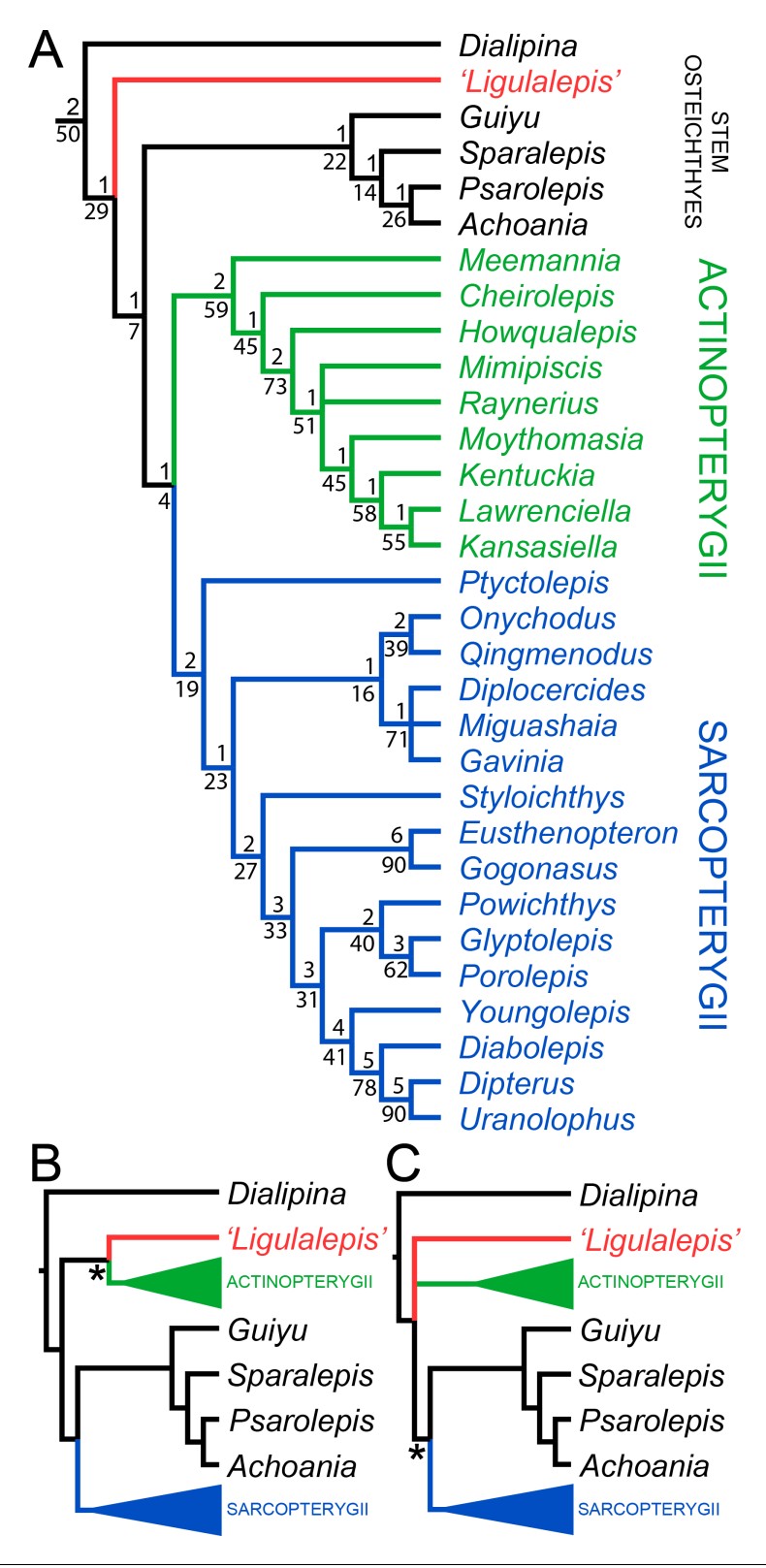

**Figure 10.** Results of parsimony phylogenetic analysis. (**A**) Strict consensus tree. Numbers above nodes refer to bremer support, numbers below nodes represent bootstrap support. (**B**) Strict consensus tree after enforcing '*Ligulalepis*' as a stem actinopterygian. (**C**) Strict consensus tree after constraining 'psarolepids' (*Guiyu, Sparalepis, Psarolepis, Achoania*) as stem sarcopterygians. Asterisks indicate constrained nodes.
DOI: https://doi.org/10.7554/eLife.34349.013

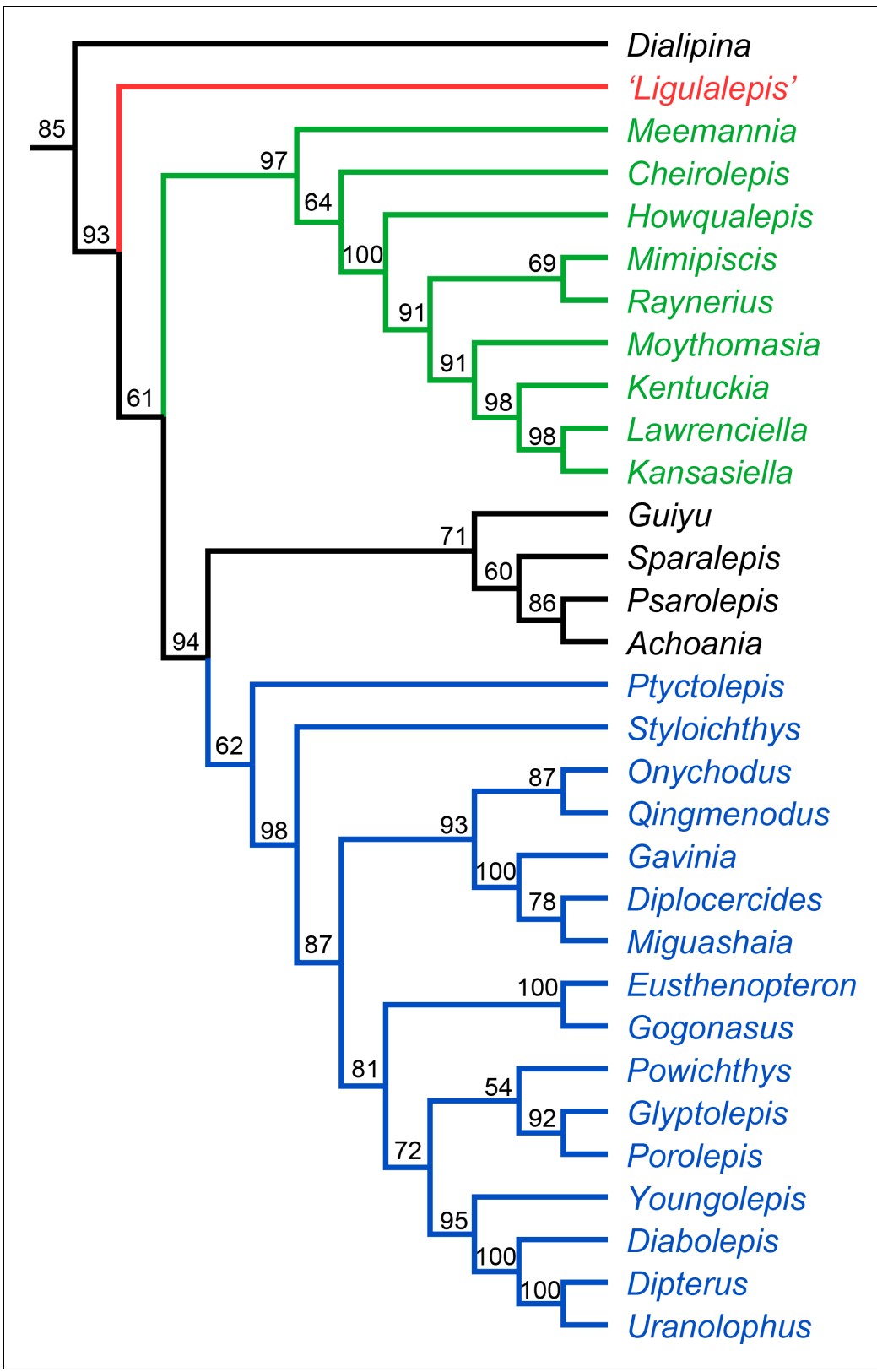

**Figure 11.** Results of Bayesian phylogenetic analysis. Maximum clade credibility tree. Numbers represent posterior probabilities, displayed as percentages for presentation purposes.

DOI: https://doi.org/10.7554/eLife.34349.014

*Giles et al., 2018*), communicate with the fossa bridgei (e.g *Pteronisculus*, *Boreosomus*: *Nielsen, 1942*; *Polyodon*: *Bridge, 1878*) or form an independent pocket (e.g. *Acipenser*, *Gardiner 1984*). *Patterson (1975)* claimed that the symmetry and even the presence of this character can vary between individuals of the same species – although investigation of several of Patterson's specimens via CT scanning has identified only symmetrical lateral cranial canals (*Giles et al., 2018*). In *Mimipiscis* the lateral cranial canal in some specimens can occupy the whole area between the posterior and anterior semicircular canals, while in others be 'little more than a pocket in front of the posterior semicircular canal' (*Gardiner, 1984*, pg. 242). *Gardiner (1984)* suggested that the lateral cranial canal can be expressed simply in terms of the degree of ossification of the dorsal otic region. The two specimens of '*Ligulalepis*' seem to confirm this idea, with the development of a lateral cranial canal variable between specimens, and the extent of the canal and perichondral lining also variable within a specimen. The endocranium of *Meemannia* is known only from a single skull specimen, so variability in development of the lateral cranial canal cannot be studied in this taxon. Mechanical preparation of *Meemannia* may also have obscured aspects of lateral cranial canal anatomy. Moreover, an actinopterygian identification for '*Ligulalepis*' is also at odds with the lack of pore canal network.

Topology tests reveal that the relationships of these early osteichthyans are somewhat interdependent, as constraining '*Ligulalepis*' to the actinopterygian stem also leads to 'psarolepids' branching from the sarcopterygian stem, necessitating independant origins of a number of characters in 'psarolepids' and non-osteichthyan gnathostomes (cf. *Lu et al., 2017*), and of tooth enamel in actinopterygians and sarcopterygians. This is because characters that support a stem osteichthyan position for '*Ligulalepis*' (i.e. the presence of an eyestalk and short olfactory tracts) are also found in *Psarolepis* and *Achoania* (*Zhu et al., 2001*; *Zhu et al., 2013*) and only support a stem osteichthyan position if all these taxa are recovered on the stem. Evidence for a stem osteichthyan position for 'psarolepids' is now accumulating, with characters such as dorsal fin spines, a median dorsal plate and absence of tooth enamel supporting this relationship (*Qu et al., 2015*; *Zhu et al., 2009*; *Lu et al., 2017*). This in turn provides additional support for a stem osteichthyan position for '*Ligulalepis*'.

In summary, our current phylogenetic hypothesis is that '*Ligulalepis*' is a stem osteichthyan. While an actinopterygian affinity requires only one extra step, this position seems to be at odds with the distribution of anatomical features amongst early osteichthyans.

## '*Ligulalepis*', histology and the problem of associated material

The skulls investigated herein are not necessarily disqualified from belonging to the same animal as the scales described for *Ligulalepis* (*Schultze, 1968*). However, we follow *Giles et al., 2015c* in maintaining the position that disassociated material cannot be unequivocally attributed to the same taxon. Scale material of *Ligulalepis* was described as actinopterygian on the basis of an anterodorsal process on the scale, 'ganoine' ridges, and a narrow scale peg (*Schultze, 1968*; *Schultze, 2016*). However, the distribution of these characters amongst osteichthyans has subsequently been comprehensively addressed by *Friedman and Brazeau (2010)*. An anterodorsal process is primitive for osteichthyans, and as the presence (and therefore relative width) of a peg cannot be assessed in outgroups the polarity of this character is ambiguous. While 'ganoine' encompasses multiple character states, some of which are general for osteichthyans (e.g. the presence of enamel, multiple layers of enamel) the presence of superimposed layers of enamel applied directly to each other is known only in actinopterygians. This indicates that *Ligulalepis*—that is the scale-based taxon—is an actinopterygian, at odds with the osteichthyan identification of '*Ligulalepis*'—that is the cranium-based taxon. A scale-based *Ligulalepis* is still problematic, however, as constituent species are erected on the basis of widespread (and often plesiomorphic characters) and span from the Ludlow of China (*Wang and Dong, 1989*) to the Emsian of Australia (*Schultze, 1968*).

The tooth and jaw fragment attributed to *Ligulalepis* recently figured by (*Schultze, 2016*, fig. 13) presents an additional problem. A vertical thin section through the tooth clearly shows an acrodin tip. Acrodin is a highly mineralized capping tissue restricted to actinopterygians crownward of *Cheirolepis* (*Friedman and Brazeau, 2010*). It is unclear which characters were used to identify this specimen as *Ligulalepis*, but it most likely does not belong to the same taxon as the skulls investigated herein. Furthermore, this tooth comes from a different fossil site (Troffs Formation, Trundle Group, Mid-Pragian-Lower Emsian of New South Wales) than the skulls described in this study. As both the

scales and jaw possess actinopterygian characters, it is possible that they belonged to the same taxon. However, in keeping with our protocol of not referring unassociated specimens (at least in the absence of clear apomorphic characters), we hesitate to support a *Ligulalepis* identity for the jaw specimen.

## Materials and methods

### Materials

This study involves the incomplete skull of '*Ligulalepis*' AM-F101607, which was previously described (*Basden and Young, 2001*; *Basden et al., 2000*), and a new specimen, ANU V3628, discovered by Ben King in late 2015. Both specimens came from the limestone outcrops on private land (Cathles' 'Cooradigbee' property) at the southern end of Goodradigbee Inlet, Wee Jasper, New South Wales, Australia. ANU V3628 was found in the Bloomfield Limestone Member of the Taemas Formation near Rocky Flat, and AM-F101607 was probably from a similar horizon, possibly at Caravan Point about 300 m to the north, although precise locality and horizon were not recorded for this specimen (although most likely from the Emsian *pireneae-serotinus* condont zone). ANU V3628 was found in a large limestone block which was trimmed with an angle grinder. The specimen was then bathed for approximately 2 hr in 5% acetic acid. The exposed bone was embedded in resin, and the block was trimmed further with an angle grinder. The specimen was then given a number of acid baths in 5% acetic acid whilst suspended upside down from a retort stand. After the skull roof became visible, further baths at progressively lower acid concentration were performed with the specimen fully immersed. Later acid baths were buffered using spent acid. Exposed bone was hardened with paraloid at intervals.

### Micro-computed tomography scanning and visualisation

AM-F101607 was scanned at the Australian National University (ANU) High Resolution Micro X-ray Computed Tomography facility (*Sakellariou et al., 2004*) with a resultant scan resolution of 30.4 microns (SI:1). ANUV3628 (SI:2) was scanned at Adelaide Microscopy on a Skyscan 1076. Specimen to source distance was 121 mm, camera to source distance was 161 mm. Source voltage was 100kV, and current 100 µA. 393 projections were taken on a Hamamatsu Orca-HRF camera. The resultant voxel size was 8.5 microns. Three-dimensional modeling and segmentation was completed using the software *VGStudio Max*, version 2.2 (Volume Graphics Inc., Germany), and *Mimics* 18.0 (Materialise Medical Co, Belgium). *Drishti* version 2.6 (*Limaye, 2012*) and Blender (blender.org; Stitching Blender Foundation, Amsterdam, the Netherlands) were also used for presentation purposes. Both CT datasets are available as Supplementary Information.

### Anatomical abbreviations

0, canal for terminal nerve 0; I, canal for olfactory nerve I; II, canal for optic nerve II; III, canal for oculomotor nerve III; IV, canal for trochlear nerve IV; V, canal for trigeminal nerve V; acv, anterior cerebral vein; ant.amp, ampulla on anterior semicircular canal; ap.f, foramen in anterior pocket; asc, anterior semicircular canal; bpt, basipterygoid process; br.buc.1, lateralis nerve branches for the dorsal part of the infraorbital canal; br.prof, canal for branches of the profundus nerve V; br.max, canals for branches of the maxillary nerve in the postnasal wall; bsp, basisphenoid; cc, crus commune; cer, space for cerebellar auricles; com.V.jug, communication between the trigeminal nerve and the jugular canal; It(Dsph), intertemporal bone (dermosphenotic of actinopterygians); die, space for the diencephalon; epsb, canal for the efferent pseudobranchial artery; esc, external semicircular canal; ext.amp, ampulla on external semicircular canal; eys, area for attachment of eyestalk; f.ica, foramen for entry of internal carotid artery; frla, foramina for ramus lateralis accessorius; g.dend, possible groove for endolymphatic duct; hmf, hyomandibular facet; hyp, space for hypophysis; hyp.v, hypophysial vein; ica, groove for internal carotid artery; ioc, postorbital branch of the infraorbital sensory line; jug.c, canal for jugular vein; lcc?, possible lateral cranial canal; mcv, canal for middle cerebral vein; mpl, middle pit line; my.IV, myodome for superior oblique eye muscle/dorsal myodome; my.III, myodome for oculomotor-innervated eye muscle; my.VI, myodome for abducens-innervated eye muscle; n.cap, nasal capsule; olf.b, space for olfactory bulb; opha, ophthalmic artery; opt.l, space for optic lobes otc otic section of the infraorbital canal; ot.lat, otic lateralis nerve branches; otc, otic canal;

"P", extension of the main sensory canal beyond infraorbital canal; Par(Fr), parietal (frontal); pcv, posterior cerebral vein; pdf, posterodorsal fontanelle; pin, pineal canal; pit, pituitary vein; por, postorbital process; PP(par), postparietal (parietal); ppl, posterior pit line; prof, canal for profundus nerve; psc, posterior semicircular canal; pv, pituitary vein; r.lat, root of the anterior lateralis nerves; s. su, sinus superior; sac, sacculus; soc, supraorbital sensory canal; soph, canal for the superficial ophthalmic nerve; sp.n, spiracular notch; St(It), supratemporal bone (intertemporal of actinopterygians); Tab(St), tabular bone (supratemporal of actinopterygians); tel, space for telencephalon; vam, ventral anterior myodome; VIIhm, canal for hyomandibular branch of the facial nerve VII; vm, ventral myodome; Vmd, canal for mandibular trunk of trigeminal nerve V; vom, area for attachment of vomer.

## Phylogenetic methods

The character matrix used was based upon the dataset of Lu et al. for their recent work on *Ptyctolepis*, which contained 278 characters and 94 taxa (*Lu et al., 2016a*). '*Ligulalepis*' was coded from the two skulls only; scale characters were not included.

Based on new information from the scans, the coding for character #31 (Sensory canals/grooves) was updated from state 0 (within thickness of skull bones) to state 1 (prominent ridges on visceral surface of skull bones). Seven other characters previously unknown in '*Ligulalepis*' were coded for the first time: #41, Pineal opening in dermal skull roof (present); #47, Number of bones of skull roof lateral to postparietals (two); #132, Canal for jugular in postorbital process (present); #152, External/horizontal semicircular canal (joins the vestibular region dorsal to posterior ampulla); #259, Position of anterior nostril (facial); #261, Three large pores associated with each side of ethmoid (absent); #263, Size of profundus canal in postnasal wall (small).

We clarified the definition of character #115 to refer only to presence or absence of dermal bone separating the nostril and orbit. Previously, the definition of this character simply referred to 'association' or 'confluence' of the nostril and the orbit, but this is not entirely satisfactory in the case of '*Ligulalepis*' where the nostril directly enters the orbit, but the dermal bones around the external opening are not completely known. A new character was introduced to reflect the different conditions of the endoskeleton around the posterior nostril. This was character #281 endoskeletal lamina (postnasal wall) separating posterior nostril and orbit: 0 (absent); 1 (present). Another new character was introduced concerning the pituitary vein, following *Castiello and Brazeau, 2018*. This was character #282 pituitary vein canal: 0 (discontinuous, enters endocranial cavity); 1 (discontinuous, enters hypophysial chamber); 2 (continuous transverse canal).

Other minor changes were #240 from one to inapplicable for *Cladoselache, Climatius* and *Cobelodus*. State 1 of character #267 (endoskeletal spiracular canal: partial enclosure or spiracular bar) was changed to (spiracular bar), to avoid grey areas as to what constitutes 'partial enclosure'. *Raynerius* was recoded as state 0 (open), and *Cheirolepis* as 0/1 (open/spiracular bar) due to uncertainty interpreting the crushed specimen (*Giles et al., 2015a*). One character (trigemino-facial recess present/absent) was deleted following *King et al., 2017*.

One skull roof character (*Lu et al., 2017*) character 43: Series of paired median skull roofing bones that meet at the dorsal midline of the skull) was reformulated into four: #277, Postparietals/centrals (0 absent/1 present); #278, Condition of postparietals/centrals (0 meet in midline/1 do not meet in midline/2 single median bone); #279, Parietals (0 absent/1 present), and #280, Condition of parietals (0 meet in midline/1 do not meet in midline).

The final matrix comprises 282 characters (see SI 3), scored for the same 94 taxa as *Lu et al., 2017*. Multistate characters were treated as unordered except for numbers 63, 125, 164, 260, 262 and 266. Parsimony analysis was performed in TNT v1.5 (*Goloboff and Catalano, 2016*). Analyses initially used new technology search for 1000 replications, using ratchet, tree fusing, sectorial search and drift search algorithms with default settings. TBR branch swapping was then performed on the resulting trees to explore the tree islands more thoroughly. A total of 1936 trees (using collapsing rule 1) of length 818 were found, and the strict consensus tree was saved. Gnathostomes (i.e. all taxa except Galeaspida and Osteostraci) were constrained to be monophyletic, and trees were rooted on Galeaspida. Bremer support values were calculated through a series of tree searches each with a negative constraint on a node in the strict consensus tree. Each of these constrained searches used the same new technology search settings as for the main analysis, for 200 replications. Bootstrap values were calculated using 1000 bootstrap replications. Within each bootstrap replication, the same new technology search settings as above were used, for 100 random addition sequence

replications. A list of apomorphies was produced using ACCTRAN for one of the shortest trees using PAUP* (*Swofford, 2003*). All scripts for all analyses are included in the supplementary information (see SI 3).

Bayesian analysis was performed in MrBayes 3.2.6 (*Ronquist et al., 2012*). The same set of characters was ordered. The MkV model (*Lewis, 2001*) was applied, with a gamma parameter to account for rate variation across characters. Four independent analyses were run (each with four chains) for 10 million generations. Convergence of the four runs was confirmed by standard deviation of split frequencies less than 0.01 and effective sample size greater than 1000 for all parameters.

## Supplementary information

The following files are available for download from DRYAD (https://doi.org/10.5061/dryad.41dh5), when using this data please cite the data package in addition to the original publication.

Supplementary Information 1: Reconstructed TIFF slices of AM-F101607.
Supplementary Information 2: Reconstructed BMP slices of ANUv3628.
Supplementary Information 3: Folder with all files for phylogenetic analysis.
Supplementary Information 4: Folder with Mimics files.

## Statement of authorship

The project was conceived by AMC and JAL. AMC, SG, BK and JAL generated the CT renderings. AMC, SG, BK, JAL and BC produced figures. GCY and BK conducted fieldwork. BK prepared and scanned one of the specimens. SG, BK, AMC and JAL conducted the phylogenetic analyses. PEA and JAL both contributed materials to the project. All authors participated in the interpretation of the specimen and writing of the manuscript.

## Acknowledgements

Thanks to Helen and Ian Cathles for access to the fossil site, Ben Young and Vincent Dupret (Australian National University) for support in the field, and Carey Burke (Flinders University) for assistance with acid preparation. We thank Tim Senden (Australian National University CT Lab) and Ruth Williams (Adelaide Microscopy) for CT scanning, and Mike Lee (Flinders University) for assistance with phylogenetic analyses. We thank Carole Burrow (Queensland Museum), Martin Rücklin (Naturalis Biodiversity Center) and Martin Brazeau (Imperial College London) for discussions about the material, and Hans-Peter Schultze for checking crucial codings for *Dialipina*. This work was supported by the Australian Research Council: JAL, AMC, GCY and BK acknowledge support from ARC DP 140101461, BC acknowledges ARC DE 160100247. SG was supported by a Junior Research Fellowship (Christ Church, Oxford) and a Royal Society Dorothy Hodgkin Research Fellowship, and PEA acknowledges the support of the Knut and Alice Wallenberg Foundation. We thank Min Zhu and two anonymous reviewers for their thorough and helpful comments on an earlier version of this manuscript.

## Additional information

### Funding

| Funder | Grant reference number | Author |
|---|---|---|
| Australian Research Council | DP 140101461 | Alice M Clement<br>Benedict King<br>Gavin C Young<br>John A Long |
| Royal Society | Dorothy Hodgkin Research Fellowship | Sam Giles |
| Australian Research Council | DE 160100247 | Brian Choo |

The funders had no role in study design, data collection and interpretation, or the decision to submit the work for publication.

## Author contributions

Alice M Clement, Conceptualization, Data curation, Formal analysis, Investigation, Methodology, Project administration, Visualization, Writing—original draft, Writing—review and editing; Benedict King, Investigation, Formal analysis, Data curation, Methodology, Visualization, Writing—original draft, Writing—review and editing; Sam Giles, Data curation, Formal analysis, Investigation, Methodology, Visualization, Writing—original draft, Writing—review and editing; Brian Choo, Visualization, Writing—review and editing; Per E Ahlberg, Resources, Software, Writing—review and editing; Gavin C Young, Resources, Writing—review and editing; John A Long, Conceptualization, Formal analysis, Resources, Software, Visualization, Writing—review and editing

## Author ORCIDs

Alice M Clement (iD) https://orcid.org/0000-0003-0380-7347
Benedict King (iD) https://orcid.org/0000-0002-9489-8274
Sam Giles (iD) https://orcid.org/0000-0001-9267-4392
John A Long (iD) https://orcid.org/0000-0001-8012-0114

## Decision letter and Author response

Decision letter https://doi.org/10.7554/eLife.34349.021
Author response https://doi.org/10.7554/eLife.34349.022

# Additional files

## Supplementary files

• Transparent reporting form
DOI: https://doi.org/10.7554/eLife.34349.015

## Data availability

The following files are available for download from DRYAD (https://doi.org/10.5061/dryad.41dh5), when using this data please cite the data package in addition to the original publication. Supplementary Information 1: Reconstructed TIFF slices of AM-F101607. Supplementary Information 2: Reconstructed BMP slices of ANUv3628. Supplementary Information 3: Folder with all files for phylogenetic analysis. Datasets Generated: Data from: Neurocranial anatomy of an enigmatic Early Devonian fish sheds light on early osteichthyan evolution: Clement A.M., 2018, doi:10.5061/dryad.41dh5, Available at Dryad Digital Repository under a CC0 Public Domain Dedication.

The following dataset was generated:

| Author(s) | Year | Dataset title | Dataset URL | Database, license, and accessibility information |
|---|---|---|---|---|
| Alice M Clement, Benedict King, Sam Giles, Brian Choo, Per E Ahlberg, Gavin C Young, John A Long | 2018 | Data from: Neurocranial anatomy of an enigmatic Early Devonian fish sheds light on early osteichthyan evolution | https://doi.org/10.5061/dryad.41dh5 | Available at Dryad Digital Repository under a CC0 Public Domain Dedication. |

The following previously published dataset was used:

| Author(s) | Year | Dataset title | Dataset URL | Database, license, and accessibility information |
|---|---|---|---|---|
| Van Essen DC, Smith SM, Barch DM, Behrens TE, Yacoub E, Ugurbil K, Consortium W–MH | 2013 | Human Connectome Project | www.humanconnectome.org | Publicly available at http://www.humanconnectome.org. |

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
