## [Decision Letter]

Thank you for submitting your article "Neurocranial anatomy of an enigmatic Early Devonian fish sheds light on early osteichthyan evolution" for consideration by *eLife*. Your article has been favorably evaluated by Diethard Tautz (Senior Editor), Min Zhu (Reviewing Editor and Reviewer #1), and two additional reviewers.

The reviewers have discussed the reviews with one another and the Reviewing Editor has drafted this decision to help you prepare a revised submission.

This is a keenly awaited manuscript to provide fresh and solid anatomical data on a critical but phylogenetically controversial taxon "*Ligulalepis*" via HXCT scanning and virtual restorations. A wealth of new neurocranial data will help to illuminate early character acquisitions of osteichthyans or bony vertebrates, which comprise some 98% of extant vertebrate species.

Summary:

For the better part of 20 years, a tiny osteichthyan braincase from the Early Devonian of Australia, which was attributed to *Ligulalepis* or "*Ligulalepis*", has proven a tantalizing enigma. Despite exceptional preservation, the specimen has been something of an 'also ran' in debates on early bony fish evolution relative to a flood of Chinese discoveries. This has little do with the importance of the Australian specimen, but instead reflects limitations in the detail/reliability of past accounts arising from the tiny size of the fossil. This contribution by Clement and colleagues, which provides detailed models of the external and internal anatomy of this original fossil plus a newly discovered specimen, is therefore a welcome addition to the body of knowledge on early bony fishes.

Overall, the anatomical results of this contribution are very useful in terms of description and figures. However, the descriptive and phylogenetic parts of the text can be greatly improved for the readability, and some issues need to be addressed (e.g. inconsistency in format, variable quality of renderings, a tendency to refer to regions of the endocast as if they were regions of the brain, and so on.). As such, we have the following suggestions and comments to improve the manuscript.

Essential revisions:

1) How to name AM-F101607 and ANU V3629 is a tricky issue. Either *Ligulalepis* or "*Ligulalepis*" is acceptable according to one reviewer, however, the authors had better use one term consistently throughout the text. The author discussed the term issue in the Introduction (sixth paragraph) and the Discussion (subsection “*Ligulalepis*, histology and the problem of associated material”) parts, which can be combined into one section.

According to the authors, *Ligulalepis* can only apply to the isolated scales if we have not found the skull specimens articulating with the scales ("we adopt the premise that dissociated skulls and scales cannot strictly be attributed to the same taxon […] with respect to *Ligulalepis*, we followed the method of Giles et al. (2015), and coded any characters relating to scale or tooth morphology as unknown).” In the meanwhile, the authors "concede it is possible that the scales do belong with the skulls as only one type of actinopterygian".

If the authors code "any characters relating to scale morphology as unknown" in the matrix, they had better use "*Ligulalepis*" to represent AM-F101607 and ANU V3629, because *Ligulalepis* was erected based on these scales. If the authors adopt the premise that the dissociated skulls and scales can be attributed to the same taxon, which is most likely based on all the available data, they can use *Ligulalepis* and code the scale morphology for this taxon. Do the histological data of AM-F101607 and ANU V3629 provide the further information to clarify the association of the skull and scale material?

2) Taxonomy and attribution: *Ligulalepis* had been considered an early actinopterygian based on isolated scales from the same and other localities, assuming it is the same taxon. This is not a solid assumption at all, as the authors explain in the Introduction; co-occurrence with isolated scales does not make the possessor of this braincase automatically the scale-bearer. This uncertainty is admitted in the manuscript – the braincases are only attributed '*Ligulalepis*' sp. In the Introduction, Materials and methods and previous work – as is the uncertainty about the relationship between different scale-based species of *Ligulalepis* from China and Australia separated by some tens of millions of years. After all, scale traits are highly functional and highly convergent in fishes; indeed, if *Ligulalepis* (scale-form) is a stem-osteichthyan, its scales would be convergent on actinopterygians. I note that the authors dismiss the attribution of a jawbone with acrodin-caped teeth (subsection “*Ligulalepis*, histology and the problem of associated material”, last paragraph) to *Ligulalepis* based on the fact that it comes from an "earlier site" without examination, but do not go that far in considering a relationship between braincase and ganoine-bearing scales which might support "ancestor" status (see below). This is inconsistent.

Yet, despite their demonstrated uncertainty about the attribution of the braincase to *Ligulalepis*, the authors suddenly drop the quotation marks around the genus name after the Introduction. The description details the neurocranial features of *Ligulalepis* full stop, and does not cover scale morphology. Their phylogenetic analysis is meant to place *Ligulalepis* as a genus but did not include scale traits. Did some of the authors conclude it is attributable after all (beyond "it cannot be disqualified" in the first paragraph of the aforementioned subsection)?

3) Phylogenetic part (subsection “Phylogenetic analyses”). There are many errors in this part. It looks that the matrix in Supplementary Information 3 (available via Dryad) was expanded from an intermediate version between Lu et al. (2016a) and Lu et al. (2017). The matrix in Lu et al. (2016a) has 269 characters (rather than 273 in the text) and 90 taxa, whiles the matrix in Lu et al. (2017) has 278 characters and 94 taxa. The matrix in Supplementary Information 3 (available via Dryad) has 280 characters (rather than 275 in the text) and 93 taxa (rather than some 90). As two analyses expanded from Lu et al. (2016a) and Lu et al. (2017) yield the same tree topology, one reviewer strongly recommend the authors to use the matrix expanded from Lu et al. (2017), which will avoid the confusions in the text.

4) Interpretation: Despite the uncertainties, the authors assert in the Abstract their main finding is that *Ligulalepis* (containing both the uncoded scale taxa from China and Australia and the braincases) is the last common ancestor of the node subtending psarolepids and osteichthyans. Leaving aside the taxonomic problems, and that this assertion is not made in the text, it is entirely unsupported and improbable. First, no ancestor sampling Bayesian analysis is presented here. A polytomy does not render one of the three clades ancestral to the others.

Second, such a result would be very unlikely given realistic priors, and basic logic, that a Devonian species from Australia is the sampled ancestor of a specific clade of Silurian osteichthyans. This would necessitate absolute stasis within *Ligulalepis* for over 20 million years, at odds with demonstrated faunal and lineage-specific changes both on a global scale and in South China during this same interval and the absence of psarolepids in the latter time. Indeed, it is even more improbable considering that the amount of morphological change observed within psarolepids over a shorter time period, and the fact that *Ligulalepis* scales changed enough over 20 million years to be identified as two different species.

This assertion also leaves aside the fact that a stem-osteichthyan position for '*Ligulalepis*' was poorly supported in parsimony analysis ("alternative placements cannot yet be ruled out"), and that its position is unresolved in the Bayesian analysis (although Figure 11 shows a branch with 44% node support separating *Ligulalepis* from crown osteichthyes, it avoids showing a polytomy as in the parsimony analysis). Therefore, it is not even clear whether '*Ligulalepis*' is a stem-osteichthyan, let alone the ur-osteichthyan. Indeed, considering that *Dialipina* was recovered as the sister group to all Osteichthyans in all parsimony analyses and the Bayesian analysis, it is more likely to be the "the ancestor" (but see above)! (Figure 10).

In summary, the Abstract and main conclusions gloss over large taxonomic uncertainty about the many forms attributed to *Ligulalepis* in order to make an unsupported point about sampled ancestry and artificially raise the impact of the manuscript. Until the authors sort out these taxonomic issues, re-evaluate support for their claims, and explain their choices more fully, it is not clear what the main result of this study is beyond the straight description and inconclusive phylogenetic analysis of a pair of braincases of uncertain affinity.

5) The 'bookends' of this paper will need some more work. Some of this, it seems, stems from *eLife*'s format of placing Materials and methods at the end of papers. Because of this, there is an abrupt transition from a semi-historical review of the fossil to gory anatomical detail to a mention of a second specimen and unspecified 'scan data'. One reviewer would strongly encourage the authors to smuggle at least some clues of what they did (not in the detail required for Materials and methods, obviously) into a rewritten final paragraph(s) of the Introduction. This will provide readers with enough of a sense of what was done to move on to the description. It would also benefit this manuscript if the authors more explicitly outlined the goals of their contribution. This is implicit in the winding narrative of past interpretations of the specimen, but it would be better to make this clear. At the other end of the paper, the Discussion seems limited. Some more nuanced comparison with past interpretations would be useful, as would some sense of the degree to which any of the new data provided for the fossils described here changes our perception of early bony fish evolution. This is touched on, but we feel like there is more here.

Finally, the Abstract needs to be substantially revised/rewritten. It is largely introductory material, and contains little of the principal results (i.e. new anatomical information) presented in the manuscript.

6) The authors described the sensory canals in two sections (subsections “Skull roof” and “Sensory canals”), which can be combined for readability.

Subsection “Skull roof”, second paragraph: where is the preopercular sensory line in Figure 1A and 1C? To be illustrated.

Subsection “Skull roof”, second paragraph: postotic canal (postotic branch of main lateral line), otic canal (otic branch of infraorbital canal), e.g. see Jarvik (1980, Figure.127). Accordingly, the otic canal is a part of the infraorbital canal. The infraorbital canal labelled by the authors is the postorbital branch of the infraorbital canal.

Subsection “Skull roof”, fourth paragraph: The author should be very careful of the nature of grooves in *Achoania* and *Psarolepis*. In both genera, the supraorbital canals run within the dermal skeleton, although the groove might have connection with the trajectory of the canal (e.g. more superficial position of the canal). CT scan of ANU V3628 should reveal whether the supraorbital canal of this segment runs as a canal or a groove on the skull roof.

7) There are many figure citation errors (e.g. subsections “Ethmoid region”, end of first paragraph; “Orbitotemporal region”, fourth and fifth paragraphs; “Ventral Surface”; “Cranial Endocast”, last paragraph; “Forebrain”, last paragraph; “Hindbrain”, third paragraph), which should be double-checked and revised. To improve the readability, one reviewer suggests the authors to cite more anatomical structures (figure abbreviations in the text).

8) Life Reconstruction: What is Figure 12 based on considering that available material consists only of isolated scales and potentially a pair of isolated braincases? The jaw material previously attributed to *Ligulalepis* was rejected in text. This image gives the false impression that the form of this fish, down to the scales and dentition, is known and characters abundant, and is therefore inappropriate. It is unlikely that this is what *Ligulalepis* looked like given the evidence at hand.

9) This contribution should be accompanied by the data files indicated in the text. These were not easily visible on the *eLife* page, and it is possible that they are not available for review without request. Because these were not provided with the review, one reviewer has not been able to assess character codes.

In terms of CT datasets, the authors should minimally provide tomograms (as either.raw/.vol files or.tiff stacks) along with surface models. See Davies et al., 2017. Open data and digital morphology. PRSB 284.

It would be nice if the authors could include some kind of simple listing of the various supplements in the text. Perhaps in Materials and methods?

---

## [Author Response]

Essential revisions:1) How to name AM-F101607 and ANU V3629 is a tricky issue. Either Ligulalepis or "Ligulalepis" is acceptable according to one reviewer, however, the authors had better use one term consistently throughout the text. The author discussed the term issue in the Introduction (sixth paragraph) and the Discussion (subsection “Ligulalepis, histology and the problem of associated material”) parts, which can be combined into one section.According to the authors, Ligulalepis can only apply to the isolated scales if we have not found the skull specimens articulating with the scales ("we adopt the premise that dissociated skulls and scales cannot strictly be attributed to the same taxon […] with respect to Ligulalepis, we followed the method of Giles et al. (2015), and coded any characters relating to scale or tooth morphology as unknown).” In the meanwhile, the authors "concede it is possible that the scales do belong with the skulls as only one type of actinopterygian".If the authors code "any characters relating to scale morphology as unknown" in the matrix, they had better use "Ligulalepis" to represent AM-F101607 and ANU V3629, because Ligulalepis was erected based on these scales. If the authors adopt the premise that the dissociated skulls and scales can be attributed to the same taxon, which is most likely based on all the available data, they can use Ligulalepis and code the scale morphology for this taxon. Do the histological data of AM-F101607 and ANU V3629 provide the further information to clarify the association of the skull and scale material?

We stand by the premise that we cannot attribute the skulls to the scale taxon without firm evidence of association, and thus refer to the two skulls investigated in this paper as “*Ligulalepis*.” This has now been consistently corrected throughout the text. We retain the considerations of this as two separate sections, but have rewritten them so that the section in the Introduction sets up the problem, and the section in the Discussion considers the problem in light of new data. The resolution of the scans is not high enough to enable detailed palaeohistological information in regards to the skull material. However, we note that dermal bone and scale histology may be radically different even within the same taxon (c.f. Cheirolepis in Lu et al., 2016).

2) Taxonomy and attribution: Ligulalepis had been considered an early actinopterygian based on isolated scales from the same and other localities, assuming it is the same taxon. This is not a solid assumption at all, as the authors explain in the Introduction; co-occurrence with isolated scales does not make the possessor of this braincase automatically the scale-bearer. This uncertainty is admitted in the manuscript – the braincases are only attributed 'Ligulalepis' sp. In the Introduction, Materials and methods and previous work – as is the uncertainty about the relationship between different scale-based species of Ligulalepis from China and Australia separated by some tens of millions of years. After all, scale traits are highly functional and highly convergent in fishes; indeed, if Ligulalepis (scale-form) is a stem-osteichthyan, its scales would be convergent on actinopterygians. I note that the authors dismiss the attribution of a jawbone with acrodin-caped teeth (subsection “Ligulalepis, histology and the problem of associated material”, last paragraph) to Ligulalepis based on the fact that it comes from an "earlier site" without examination, but do not go that far in considering a relationship between braincase and ganoine-bearing scales which might support "ancestor" status (see below). This is inconsistent.Yet, despite their demonstrated uncertainty about the attribution of the braincase to Ligulalepis, the authors suddenly drop the quotation marks around the genus name after the Introduction. The description details the neurocranial features of Ligulalepis full stop, and does not cover scale morphology. Their phylogenetic analysis is meant to place Ligulalepis as a genus but did not include scale traits. Did some of the authors conclude it is attributable after all (beyond "it cannot be disqualified" in the first paragraph of the aforementioned subsection)?

As per our response to 1), we stand by the premise that we cannot attribute the skulls to the scale taxon without evidence of association, and thus refer to the two skulls investigated in this paper as “*Ligulalepis*”. We have edited the text so that this usage is consistent throughout. *Ligulalepis* thus remains a scale-based taxon, albeit one that is highly problematic. We have added more detailed consideration of potential relationships between the scales, jaw and skulls to the Discussion, noting that given the osteichthyan-like characters of the skulls and the actinopterygian-like characters indicate that the scales and skulls do not belong to the same taxon (although the scales and jaw may well do).

3) Phylogenetic part (subsection “Phylogenetic analyses”). There are many errors in this part. It looks that the matrix in Supplementary Information 3 (available via Dryad) was expanded from an intermediate version between Lu et al. (2016a) and Lu et al. (2017). The matrix in Lu et al. (2016a) has 269 characters (rather than 273 in the text) and 90 taxa, whiles the matrix in Lu et al. (2017) has 278 characters and 94 taxa. The matrix in Supplementary Information 3 (available via Dryad) has 280 characters (rather than 275 in the text) and 93 taxa (rather than some 90). As two analyses expanded from Lu et al. (2016a) and Lu et al. (2017) yield the same tree topology, one reviewer strongly recommend the authors to use the matrix expanded from Lu et al. (2017), which will avoid the confusions in the text.

We have now run all the analyses on a single matrix, an updated version of Lu et al. 2017 (for *Ptyctolepis*). Confusion regarding number of characters etc. has been fixed.

4) Interpretation: Despite the uncertainties, the authors assert in the Abstract their main finding is that Ligulalepis (containing both the uncoded scale taxa from China and Australia and the braincases) is the last common ancestor of the node subtending psarolepids and osteichthyans. Leaving aside the taxonomic problems, and that this assertion is not made in the text, it is entirely unsupported and improbable. First, no ancestor sampling Bayesian analysis is presented here. A polytomy does not render one of the three clades ancestral to the others.Second, such a result would be very unlikely given realistic priors, and basic logic, that a Devonian species from Australia is the sampled ancestor of a specific clade of Silurian osteichthyans. This would necessitate absolute stasis within Ligulalepis for over 20 million years, at odds with demonstrated faunal and lineage-specific changes both on a global scale and in South China during this same interval and the absence of psarolepids in the latter time. Indeed, it is even more improbable considering that the amount of morphological change observed within psarolepids over a shorter time period, and the fact that Ligulalepis scales changed enough over 20 million years to be identified as two different species.This assertion also leaves aside the fact that a stem-osteichthyan position for 'Ligulalepis' was poorly supported in parsimony analysis ("alternative placements cannot yet be ruled out"), and that its position unresolved in the Bayesian analysis (although Figure 11 shows a branch with 44% node support separating Lingulalepis from crown osteichthyes, it avoids showing a polytomy as in the parsimony analysis). Therefore, it is not even clear whether 'Ligulalepis' is a stem-osteichthyan, let alone the ur-osteichthyan. Indeed, considering that Dialipina was recovered as the sister group to all Osteichthyans in all parsimony analyses and the Bayesian analysis, it is more likely to be the "the ancestor" (but see above)! (Figure 10).In summary, the Abstract and main conclusions gloss over large taxonomic uncertainty about the many forms attributed to Ligulalepis in order to make an unsupported point about sampled ancestry and artificially raise the impact of the manuscript. Until the authors sort out these taxonomic issues, re-evaluate support for their claims, and explain their choices more fully, it is not clear what the main result of this study is beyond the straight description and inconclusive phylogenetic analysis of a pair of braincases of uncertain affinity.

This was a simple oversight, and we apologise for the confusion. We meant that “*Ligulalepis*” was resolved as a stem osteichthyan, specifically as the sister taxon to the ‘psarolepids’ plus crown osteichthyans.

5) The 'bookends' of this paper will need some more work. Some of this, it seems, stems from eLife's format of placing Materials and methods at the end of papers. Because of this, there is an abrupt transition from a semi-historical review of the fossil to gory anatomical detail to a mention of a second specimen and unspecified 'scan data'. One reviewer would strongly encourage the authors to smuggle at least some clues of what they did (not in the detail required for Materials and methods, obviously) into a rewritten final paragraph(s) of the Introduction. This will provide readers with enough of a sense of what was done to move on to the description. It would also benefit this manuscript if the authors more explicitly outlined the goals of their contribution. This is implicit in the winding narrative of past interpretations of the specimen, but it would be better to make this clear. At the other end of the paper, the Discussion seems limited. Some more nuanced comparison with past interpretations would be useful, as would some sense of the degree to which any of the new data provided for the fossils described here changes our perception of early bony fish evolution. This is touched on, but we feel like there is more here.Finally, the Abstract needs to be substantially revised/rewritten. It is largely introductory material, and contains little of the principal results (i.e. new anatomical information) presented in the manuscript.

We have substantially rewritten the Introduction and Discussion to address these problems. In particular, we have amended the final paragraph of the Introduction to include more detail of the methods and an explicit stamen of the goals of the manuscript. The Discussion has been re-organised to reflect this, and has also been expanded. We have also heavily revised the Abstract to include more details of new anatomical information and how interpretations have changed.

6) The authors described the sensory canals in two sections (subsections “Skull roof” and “Sensory canals”), which can be combined for readability.Subsection “Skull roof”, second paragraph: where is the preopercular sensory line in Figure 1A and 1C? To be illustrated.Subsection “Skull roof”, second paragraph: postotic canal (postotic branch of main lateral line), otic canal (otic branch of infraorbital canal), e.g. see Jarvik (1980, Figure 127). Accordingly, the otic canal is a part of the infraorbital canal. The infraorbital canal labelled by the authors is the postorbital branch of the infraorbital canal.Subsection “Skull roof”, fourth paragraph: The author should be very careful of the nature of grooves in Achoania and Psarolepis. In both genera, the supraorbital canals run within the dermal skeleton, although the groove might have connection with the trajectory of the canal (e.g. more superficial position of the canal). CT scan of ANU V3628 should reveal whether the supraorbital canal of this segment runs as a canal or a groove on the skull roof.

The two sections describing the sensory canals have now been combined. The reviewers suggest using the convention of referring to the otic and infraorbital canals as branches of the infraorbital, following Jarvik 1980. However, we note that other conventions are possible including the one we followed e.g. Northcutt 1989, ZFIN anatomy database etc. Arguably, it is something of an arbitrary choice. However, the otic canal develops from a separate placode from the infraorbital and supraorbital canals (e.g. Northcutt 1997 Brain Behav Evol 50:25-37). It is better for naming conventions to follow the underlying development in our opinion. It also has the advantage of shorter, clearer names, and avoids the anomaly of referring to a canal on top of the head posterior to the orbit as “infraorbital”. There is no notch for the preopercular sensory line, and this is now reflected in the manuscript text. We have re-examined the supraorbital canals in our specimens and compared them to *Achoania* and *Psarolepis* and are confident the structure is the same: the canals are carried in a canal on the visceral surface of the skull roof in ‘*Ligulalepis*’, but the canal is open dorsally to the skull roof. This appears to be identical to the condition in *Psarolepis* and *Achoania*, as far as can be judged form published descriptions.

7) There are many figure citation errors (e.g. subsections “Ethmoid region”, end of first paragraph; “Orbitotemporal region”, fourth and fifth paragraphs; “Ventral Surface”; “Cranial Endocast”, last paragraph; “Forebrain”, last paragraph; “Hindbrain”, third paragraph), which should be double-checked and revised. To improve the readability, one reviewer suggests the authors to cite more anatomical structures (figure abbreviations in the text).

All figure citations have been double-checked and anatomical structures of discussion are now listed within the figure citation.

8) Life Reconstruction: What is Figure 12 based on considering that available material consists only of isolated scales and potentially a pair of isolated braincases? The jaw material previously attributed to Ligulalepis was rejected in text. This image gives the false impression that the form of this fish, down to the scales and dentition, is known and characters abundant, and is therefore inappropriate. It is unlikely that this is what Ligulalepis looked like given the evidence at hand.

Figure 12 is based on the morphology of the two skulls investigated within the manuscript. The other features are hypothetical reconstructions. The figure label has been updated to “Life reconstruction of *Ligulalepis* based on the skull morphology of AM-F101607 and ANU V3628, other features remain hypothetical” to more clearly express this.

9) This contribution should be accompanied by the data files indicated in the text. These were not easily visible on the eLife page, and it is possible that they are not available for review without request. Because these were not provided with the review, one reviewer has not been able to assess character codes.In terms of CT datasets, the authors should minimally provide tomograms (as either.raw/.vol files or.tiff stacks) along with surface models. See Davies et al., 2017. Open data and digital morphology. PRSB 284.It would be nice if the authors could include some kind of simple listing of the various supplements in the text. Perhaps in Materials and methods?

All data files were included in the submission to *eLife* and it was the understanding of the authors that all would be available for review. Reconstructed tomogram (BMP or TIFF) stacks were provided as Supplementary Information for both specimens (Supplementary Information 1 and 2). A listing of all the Supplementary Information files and their contents has now been added at the end of the Materials and methods section. All Supplementary Information files have been uploaded to DRYAD and can be accessed via:

https://doi.org/10.5061/dryad.41dh5